# Heterogeneous integration of single-crystalline rutile nanomembranes with steep phase transition on silicon substrates

Dong Kyu Lee[1,4], Yunkyu Park[1,4], Hyeji Sim [1,4], Jinheon Park[1], Younghak Kim[2], Gi-Yeop Kim[1], Chang-Beom Eom[3], Si-Young Choi [1✉] & Junwoo Son [1✉]

Unrestricted integration of single-crystal oxide films on arbitrary substrates has been of great interest to exploit emerging phenomena from transition metal oxides for practical applications. Here, we demonstrate the release and transfer of a freestanding single-crystalline rutile oxide nanomembranes to serve as an epitaxial template for heterogeneous integration of correlated oxides on dissimilar substrates. By selective oxidation and dissolution of sacrificial $VO_2$ buffer layers from $TiO_2/VO_2/TiO_2$ by $H_2O_2$, millimeter-size $TiO_2$ single-crystalline layers are integrated on silicon without any deterioration. After subsequent $VO_2$ epitaxial growth on the transferred $TiO_2$ nanomembranes, we create artificial single-crystalline oxide/ Si heterostructures with excellent sharpness of metal-insulator transition ($\triangle\rho/\rho > 10^3$) even in ultrathin (<10 nm) $VO_2$ films that are not achievable via direct growth on Si. This discovery offers a synthetic strategy to release the new single-crystalline oxide nanomembranes and an integration scheme to exploit emergent functionality from epitaxial oxide heterostructures in mature silicon devices.

---

[1] Department of Materials Science and Engineering, Pohang University of Science and Technology (POSTECH), Pohang, Republic of Korea. [2] Pohang Accelerator Laboratory, Pohang, Republic of Korea. [3] Department of Materials Science and Engineering, University of Wisconsin-Madison, Madison, WI, USA. [4]These authors contributed equally: Dong Kyu Lee, Yunkyu Park, Hyeji Sim. ✉email: youngchoi@postech.ac.kr; jwson@postech.ac.kr

Heteroepitaxial growth has been widely used to obtain single-crystal films for developing modern solid-state electronic and photonic devices[1,2]. In particular, epitaxial oxide heterostructures provide promise for emerging electronics and photonics due to the intriguing phenomena (e.g., metal-insulator transition (MIT)[3–6], high-electron mobility[7], ferroelectricity[8]) exhibited by these ionic crystals. Therefore, the stacking of single-crystalline oxide films on dissimilar substrates (e.g., silicon (Si)) will offer ways to integrate the emergent phenomena of oxides with mature electronic and photonic devices[9–13]. However, the heteroepitaxial growth of oxide films drastically limits the possible material combinations due to the requirement of lattice matching between the epilayer and substrate[14]. For instance, direct oxide growth on dissimilar materials typically forms defective or polycrystalline layers near the interface between films and substrates[15,16], preventing unrestricted integration of single-crystal oxide films onto any desired substrates, especially on mainstream Si substrates.

Release and transfer of freestanding single-crystal sheets with a nanoscale thickness (i.e., epitaxial lift-off for nanomembrane (NM)) gives the freedom to transfer the released epilayer onto highly mismatched or amorphous substrates, and even allows reusable substrates[9,17–21]. While crystalline materials that are intrinsically layered (e.g. two-dimensional (2D) materials) are exfoliated spontaneously due to weak van der Waals bonding between layers[22], the freestanding NM from three-dimensional (3D) oxide crystals with strong bonding is hindered by the technical challenges of lifting strongly bonded epitaxial films from the oxide substrates. Thus, the technique for releasing the freestanding NM from a host substrate essentially requires the preferential creation of bond breaking from the substrate.

Many techniques have been employed to form freestanding NMs with a mechanically cleavable plane. For example, physical release methods (e.g., laser lift-off[20]) were originally developed to release epitaxial GaN semiconductor films to break strong bonding. However, these methods are only applicable to the formation of thick semiconductor membranes due to the inevitable structural damage. Moreover, a few monolayers of graphene could be inserted to release single-crystalline oxide membranes from substrates (i.e., "remote" epitaxy). Despite the versatility of this technique, the coalescence of localized nuclei in oxide layers on the graphene, along with the restriction of oxygen environment during growth, prevents the layer-by-layer growth of ultrathin oxide NM with atomic precision and high quality[10].

By contrast, the freestanding NM was chemically released from the substrate by selective etching of sacrificial layers[18,19,21]; these chemical lift-off methods are less destructive than physical methods. However, the harsh wet condition with a strong acid or base etchant typically leaves roughening and residue on the host substrates or released membrane after the chemical etch[23,24]. Recently, atomically thin perovskite oxide NM was gently released by dissolving water-soluble $Sr_3Al_2O_6$ sacrificial layers[11], but the moisture-sensitive nature of these layers prevents long-time exposure of the sacrificial layers, which restricts practical application for heterogeneous integration of oxide NM. Moreover, the development of oxide NM has been limited for perovskite structure among chemical lift-off methods so far[10–12,21,25,26]; to extend the materials spectrum for freestanding oxide NM, a new combination of the sacrificial layer and etchant needs to be developed for the heterogeneous integration of epitaxial oxide NM with other crystal structures on dissimilar substrates[10].

Here, we demonstrate that single-crystalline rutile oxide NM with the sharpened MIT can be integrated on the technologically influential, but more challenging, Si substrates by new epitaxial lift-off combination (see Fig. 1 for the process schematic). After the synthesis of an epitaxial $TiO_2/VO_2$ heterostructure on $TiO_2$ host substrate (Fig. 1a), the $VO_2$ sacrificial layer is selectively dissolved in dilute $H_2O_2$ to release the top $TiO_2$ film with the mechanical supporting layers (Fig. 1b). Contrary to the previous chemical lift-off using extreme pH solution, dilute aqueous $H_2O_2$ with mild pH leads to selective etching of epitaxially grown $VO_2$ sacrificial films by phase transformation to two-dimensional layered structure with weak bonding; this selective dissolution of $VO_2$ layer releases millimeter-scale freestanding $TiO_2$ NM from the $TiO_2/VO_2/TiO_2$ heterostructures at room temperature. Then, the single-crystalline $TiO_2$ NM is transferred onto the Si substrates without any deterioration of crystal quality (Fig. 1c, d). Interestingly, transferred $TiO_2$ single-crystal NM serves as a template for the heterogeneous integration of single-crystal $VO_2$ films on Si substrate (Fig. 1e). As a result of the $VO_2$ film epitaxially grown on $TiO_2$ NM template, more than three orders of

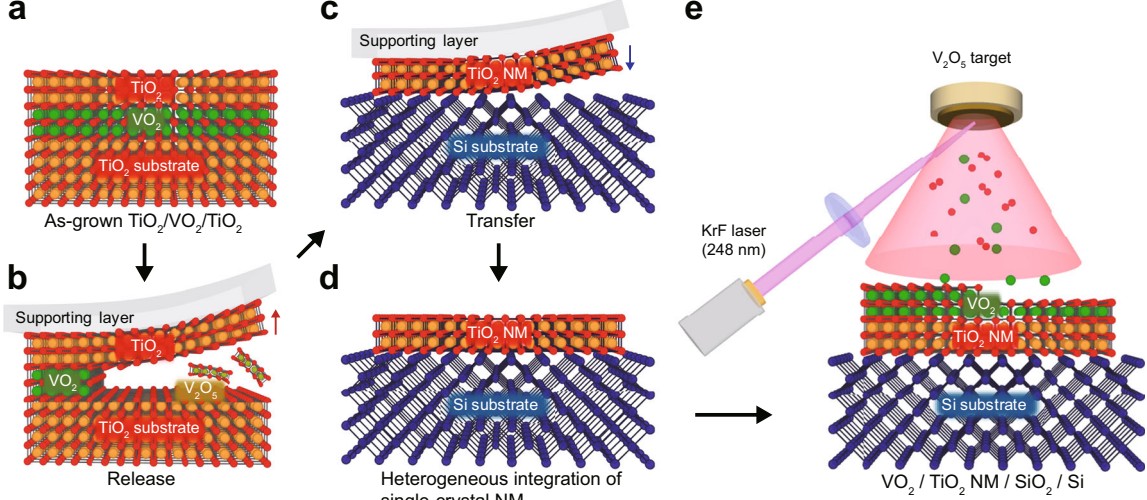

**Fig. 1 Process schematic for single-crystal rutile oxide nanomembranes (NM) on silicon. a** Schematic of an epitaxial $TiO_2/VO_2$ heterostructure on $TiO_2$ host substrate. **b** The $VO_2$ layer is dissolved in $H_2O_2$ to release the top $TiO_2$ film with the mechanical supporting layer. **c** The freestanding $TiO_2$ NM is transferred onto the desired substrates (e.g., silicon). **d** By removing the rigid supporting layer, single-crystalline rutile oxide NM is heterogeneously integrated into a silicon substrate. **e** Epitaxial $VO_2$ film with steep phase transition is grown on the $TiO_2$-NM-templated Si substrates.

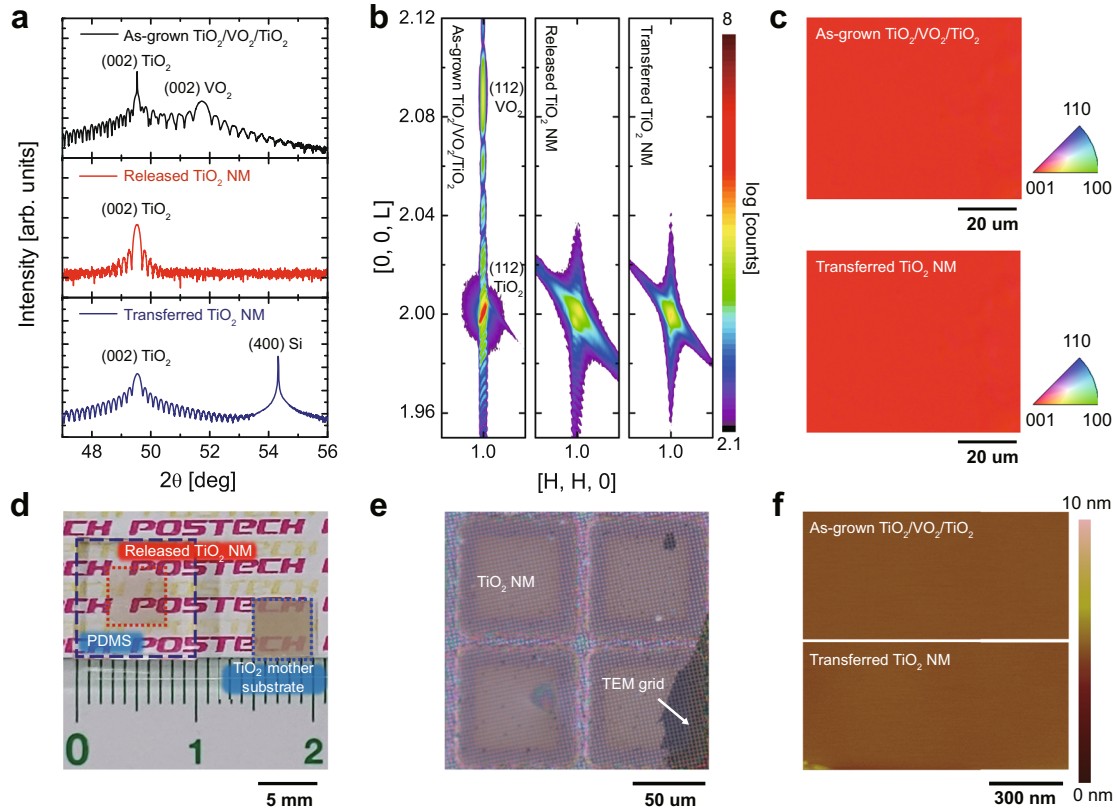

**Fig. 2 Crystal and surface quality of released and transferred TiO₂ single-crystal nanomembrane (NM).** **a** Symmetric $2\theta - \omega$ scan. **b** Reciprocal space mapping around (112) reflection of as-grown TiO₂ epitaxial films and TiO₂ single-crystal NM released on supporting layer and transferred on Si substrates. The high-resolution X-ray diffraction measurements show single-crystallinity with uniform out-of-plane orientation and no in-plane rotation in released and transferred TiO₂ NM. **c** EBSD maps of the as-grown TiO₂ epitaxial films on VO₂/TiO₂ substrates (top) and transferred TiO₂ NM on Si substrates (bottom), confirming the single-crystalline out-of-plane orientation. **d** Photograph of 40-nm-thick TiO₂ single-crystal NM released on mechanical support with the identical lateral dimension with TiO₂ host substrates. **e** OM image of transferred TiO₂ NM on the carbon TEM grid. **f** AFM image of as-grown TiO₂ films and transferred TiO₂ NM, which confirms a uniform and flat surface without surface cracks or residues.

magnitude modulation of resistivity ratio ($\triangle\rho/\rho > 10^3$) and sharpened MIT (i.e., narrow FWHM of Gaussian fitting from d $(\log_{10}(\rho))/dT$ during heating and cooling ($\triangle T_h$ and $\triangle T_c$ ~3 K)) are remarkably achieved even in the ultrathin (≤10 nm) VO₂ films on Si substrates, which is not possible using conventional thin film growth.

## Results

**Synthesis of freestanding single-crystalline TiO₂ nanomembranes.** Prior to release and transfer of TiO₂ NM, TiO₂ (10–70 nm)/VO₂ (~17 nm) heterostructures were epitaxially grown on (001)-oriented TiO₂ substrates by pulsed laser deposition (PLD) (Fig. 1a). Symmetrical $2\theta$-ω scan using synchrotron x-ray scattering on TiO₂/VO₂/TiO₂ detected two $(002)_R$ Bragg reflections, one from the rutile TiO₂ ($2\theta = 49.54°$) film and substrate, and one from VO₂ film ($2\theta = 51.75°$) (black line in Fig. 2a)[27]. Due to the close resemblance in crystal structure and small in-plane lattice mismatch of $(001)_R$-VO₂ and (001)-TiO₂ ($f_a \approx 0.86\%$ along a- and b-axis), 17-nm-thick $(001)_R$-VO₂ films are fully strained by biaxial tensile strain; all heterostructure maintained identical in-plane lattice constant (left figure of Fig. 2b). Thus, the thickness of VO₂ (~17 nm) was determined to create unstrained and defect-free TiO₂ epitaxial films regardless of film thickness[27,28].

Then, rigid support layer (e.g., PDMS) was coated on the heterostructure before release to mechanically stabilize the TiO₂ NM and to facilitate the subsequent transfer of the released

NM[10–12,25]. By simply immersing these TiO₂/VO₂/TiO₂ heterostructures into the dilute (10%) hydrogen peroxide (H₂O₂) solution at room temperature (Fig. 1b), a few millimeter-scale TiO₂ films were successfully released from their substrates by fully dissolving VO₂ sacrificial layers (i.e., denoted as released TiO₂ NM), allowing the growth substrate to be removed. The released single-crystal NM was transferred on any arbitrary substrate (i.e., denoted as transferred TiO₂ NM), including on SiO₂-coated (~80 nm) Si (100) substrates (Fig. 1c). The transferred NM was then heated at 60 °C to form stronger adhesion with substrates and the support layer was slowly detached (Fig. 1d).

Interestingly, the VO₂ sacrificial layer was dissolved selectively and rapidly in the dilute H₂O₂ solution with mild pH (~5.3) (Fig. 1b). The combination of the VO₂ sacrificial layer and H₂O₂ etchant for single-crystal rutile oxide NM in terms of release rate is as effective as that of a Sr₃Al₂O₆ sacrificial layer and H₂O for a single-crystal perovskite oxide NM[11]. The high etching capability of H₂O₂ on VO₂ was confirmed by direct comparison with that of an HNO₃ solution with the same concentration as a well-established VO₂ etchant with a strong acid (pH ~2)[24] (Supplementary Fig. 1): While the residue of VO₂ films remained even after 64 s after VO₂ films were immersed into strong acid HNO₃ solution, the H₂O₂ solution completely removed VO₂ films that were free of any residue on the surface of the TiO₂ substrate within 4 s. Furthermore, repeating H₂O₂ etching preserves the atomically flat surface topography, implying substrate reusability for the production of TiO₂ NM (Supplementary Fig. 2). VO₂ films

could be spontaneously oxidized by $H_2O_2$ and transformed to $V_2O_5$ and/or water-soluble $V_2O_5 \cdot nH_2O(s)$ gels with a layered van der Waals (vdW) structure along the c-axis[29,30]; these layered crystals with weak bonding are exfoliated and dispersed in the solution (Supplementary Fig. 3a, b). Since unreacted $TiO_2$ epitaxial layers cover whole surfaces of $VO_2$ sacrificial layers during the release process, the release of epitaxial $TiO_2$ NM by oxidation and dissolution of $VO_2$ sacrificial layers begins from the edge of the substrate (i.e., side etching); the dissolution (or release) time increases with the lateral size of the $TiO_2$ substrate (Supplementary Fig. 4).

As a result of effective dissolution of the $VO_2$ sacrificial layer by $H_2O_2$, the quality and alignment of epitaxial $TiO_2$ layers are intact during the release and transfer process as observed in symmetric 2θ-ω scans using synchrotron X-ray scattering (Fig. 2a). While the $(002)_R$ peak from the $VO_2$ sacrificial layer (2θ = 51.75°) was clearly removed after the selective etching process (red line in Fig. 2a), the $(002)_R$ peak from ~ 40-nm-thick (001) $TiO_2$ single-crystal films with thickness oscillations was observed consistently at 2θ = 49.54° in released (red line in Fig. 2a) and transferred $TiO_2$ NM on the Si substrate (blue line in Fig. 2a and Supplementary Fig. 5 with Cu $K_{\alpha 1}$ X-ray radiation). Moreover, the $(002)_R$ peak from $TiO_2$ NM simultaneously appeared with a (400) peak from the Si substrate (2θ = 54.32°) in the transferred $TiO_2$ NM (blue line in Fig. 2a), which represents that the c-axis of single-crystal $TiO_2$ NM was aligned with the out-of-plane orientation of the Si substrate. Azimuthal X-ray diffraction (XRD) φ scanning of $(112)_R$ crystallographic plane showed identical fourfold symmetry of released and transferred $TiO_2$ NMs with those of the single-crystal (001) $TiO_2$ host substrate (Supplementary Fig. 6); this result showed in-plane single-crystallinity of NMs without any rotated domains. Additionally,

an electron backscatter diffraction (EBSD) map confirmed (001) orientation of rutile crystals over a large area in both as-grown and transferred $TiO_2$ layers (Fig. 2c).

To obtain more detailed information on the crystal structures of $TiO_2$ NMs during the release and transfer process, both in-plane and out-of-plane lattice parameters were monitored using reciprocal space mapping (RSM) around the (112) reflection of as-grown, released, and transferred 70-nm-thick $TiO_2$ layers (Fig. 2b). The RSM data clearly show sharp and intense (112) Bragg reflections and Kiessig fringes from the $TiO_2$ layer and substrates, and from the $VO_2$ sacrificial layers in as-grown heterostructures. Since 17-nm-thick $VO_2$ sacrificial layers are coherently grown on the $TiO_2$ substrates with identical H (i.e., in-plane reciprocal space unit), strain-free epitaxial $TiO_2$ layers with various thickness (10–70 nm) are coherently grown on $VO_2/TiO_2$ substrates (Fig. 2b and Supplementary Fig. 7b, c). After selective etching of the $VO_2$ sacrificial layer, H and L in released and transferred $TiO_2$ NM was identical with those in $TiO_2$ substrates and films in as-grown heterostructures; these strain-free $TiO_2$ layers with precisely controlled thickness are released and transferred onto a Si substrate without modification of the lattice parameters and crystallinity (Supplementary Fig. 7a).

Furthermore, surfaces of the released and transferred $TiO_2$ NM are uniform and intact without any defective boundaries[31]. Figure 2d shows that the entire area of $TiO_2$ films with lateral dimension of millimeter-scale and nanometer thickness was fully released from the $VO_2/TiO_2$ substrates after this etching process. Optical microscope (OM) images of transferred $TiO_2$ NM on the carbon TEM grid exhibit crack-free layers with natural wrinkles (Fig. 2e and Supplementary Fig. 8a). Both scanning electron microscope (SEM, Supplementary Fig. 8b) and atomic force microscope (AFM, Fig. 2f) images confirmed

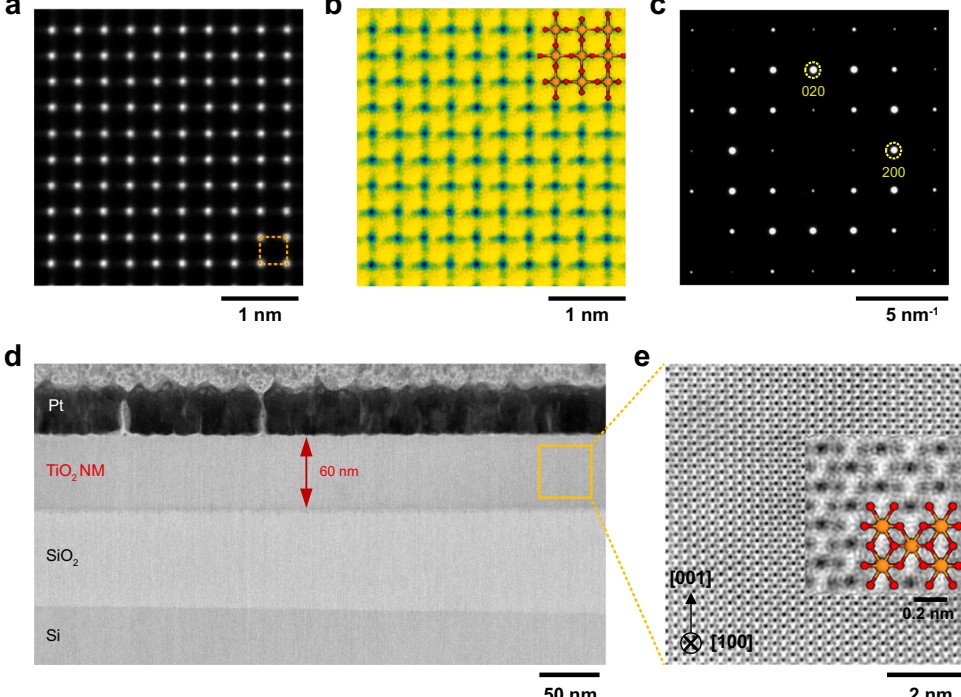

**Fig. 3 STEM analysis of freestanding single-crystal $TiO_2$ nanomembrane (NM). a** HAADF-STEM, **b** ABF-STEM, and **c** SADP images from the plane-view observation on 20-nm-thick freestanding $TiO_2$ NM. The white spots in HAADF-STEM, which are identical to Ti atoms, are consistent with cation sites of $[001]_R$-projected rutile structure. The atomic arrangement of ABF-STEM, visualization of oxygen atoms, was also identical to anion sites of rutile structure; the results indicate that atomic arrangement of freestanding $TiO_2$ NM was identical to the single-crystal rutile structure without defective features. **d, e** are the cross-sectional ABF-STEM images of $TiO_2$ NM, the quality of which is crystallographically high with perfect registry of Ti (orange color in **e**) and O (red color in **e**) atoms in rutile structure.

uniform and flat surface of both released and transferred $TiO_2$ NM without surface cracks or residues: the arithmetic average surface roughness of transferred $TiO_2$ NM ($R_a = 0.06$ nm) was comparable to that of originally grown $TiO_2$ films on $VO_2/TiO_2$ substrates ($R_a = 0.04$ nm), representing residue-free $VO_2$ removal by $H_2O_2$ etching and subsequent surface cleaning (Fig. 2f and Supplementary Figs. 9, 10).

High-resolution scanning transmission electron microscopy (STEM) confirms the local atomic structure of freestanding single-crystal $TiO_2$ NM. First, the released $TiO_2$ NM was transferred onto a TEM grid for STEM observation (Fig. 2e and Supplementary Fig. 11); the high-angle annular dark-field (HAADF) and the annular bright-field (ABF) STEM were applied for the plane-view observation. The HAADF-STEM image shows a square pattern of four titanium atoms (orange, dotted square) without a lattice distortion or defects (Fig. 3a). More interestingly, the ABF-STEM image shows the contrast tails from the titanium atom with contribution by the oxygen atoms and the pattern exactly matched the atomic structure of the rutile $TiO_2$ with the [001] zone axis (Fig. 3b). The selected-area diffraction pattern (SADP) provided that the in-plane lattice parameters of $TiO_2$ NM were 0.46 nm (Fig. 3c), which confirms the freestanding $TiO_2$ film was perfectly transferred. After the $TiO_2$ NM (~60 nm) was transferred to the $SiO_2/Si$ substrate, cross-sectional STEM observation was performed with the [100] zone axis to visualize the structural coherency of the freestanding oxide NM, as shown in Fig. 3d, e. The low magnification ABF-STEM image shows that the $TiO_2$ NM is free of defects even after its transfer process from the wide field of view (Fig. 3d); the high magnification ABF-STEM image again verifies the perfect registry of Ti (orange) and oxygen (red) atoms in the rutile structure (Fig. 3e).

**Heterogeneous integration of single-crystalline $VO_2$ films on the $TiO_2$-NM-templated Si substrates.** After transfer to the ($SiO_2$-coated) Si substrates, the single-crystalline $TiO_2$ NMs served as templates for the epitaxial growth of high-quality $VO_2$ thin films on Si substrates (Fig. 1e). As observed in symmetric $2\theta – \omega$ scans, the intense $(002)_R$ $VO_2$ peak appeared at $\sim 2\theta = 51.85°$ along with peaks related to the $TiO_2$ template ($\sim 2\theta = 49.54°$) and Si substrates ($\sim 2\theta = 54.34°$) for the $VO_2$ films on the $TiO_2$-templated Si substrates (red line in Fig. 4a), compared to the absence of related peaks for the $VO_2$ films directly grown on Si substrate (black line in Fig. 4a); this result reveals that lattice-matched single-crystal templates facilitate the formation of epitaxial $VO_2$ films on Si substrates. Single-crystallinity of $VO_2$ films on $TiO_2$ NM/Si was again verified by identical fourfold symmetry with $TiO_2$ NM templates in asymmetric $\phi$-scans of the $(112)_R$ plane (Fig. 4b). While the out-of-plane strain state of the $TiO_2$ NM did not change after the growth of $VO_2$ films, the out-of-plane lattice parameters of $VO_2$ films (2.838 Å) were reduced compared to those of bulk $VO_2$ (2.88 Å). Indeed, an RSM near the (112) reflection of the $TiO_2$ NM confirms that the peaks from the $VO_2$ films and $TiO_2$ NM showed identical H (i.e., in-plane reciprocal space unit) (Fig. 4c), which indicates that a 10-nm-thick $VO_2$ film remains coherently strained to the $TiO_2$ NM along the in-plane direction[27,28].

Low magnification ABF- and high magnification HAADF-STEM images (Fig. 4d, e) confirm that the epitaxial growth of 10-nm-thick $VO_2$ on $TiO_2$ NM can realize heterogeneous integration of single-crystal $VO_2$ films on a Si substrate. In particular, $VO_2$ and $TiO_2$ are coherently matched and the interface can be visualized due to the slight contrast difference between $VO_2$ and $TiO_2$, as indicated by the arrow (Fig. 4e); the atomic resolution image implicates that the $VO_2$ layer is tightly constrained from the underlying $TiO_2$ NM, and thus

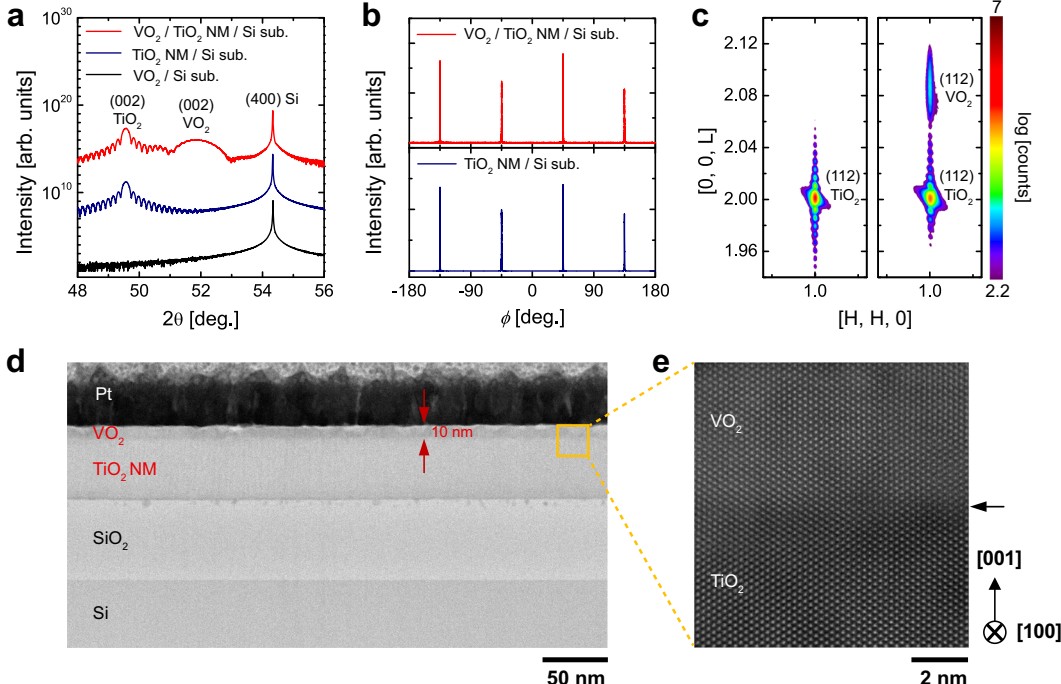

**Fig. 4 Heterogeneous integration of rutile oxide epitaxial $VO_2/TiO_2$ heterostructures on Si substrates.** **a** Symmetric $2\theta − \omega$ scan, **b** asymmetric $\phi$ scan, and **c** reciprocal space mapping around (112) reflection of $TiO_2$ NM on Si substrates before (blue in **a**, **b**, and left in **c**) and after (red in **a**, **b**, and right in **c**) the growth of $VO_2$ films, implying single-crystallinity of coherently strained $VO_2$ films on $TiO_2$ NM/Si. **d** The cross-sectional ABF-STEM images of the heterogeneous $VO_2/TiO_2$ NM on $SiO_2/Si$. The atomic-scale HAADF-STEM image at the area indicated by the yellow square in **d** is shown in **e** (zone axis: [100] in $TiO_2$).

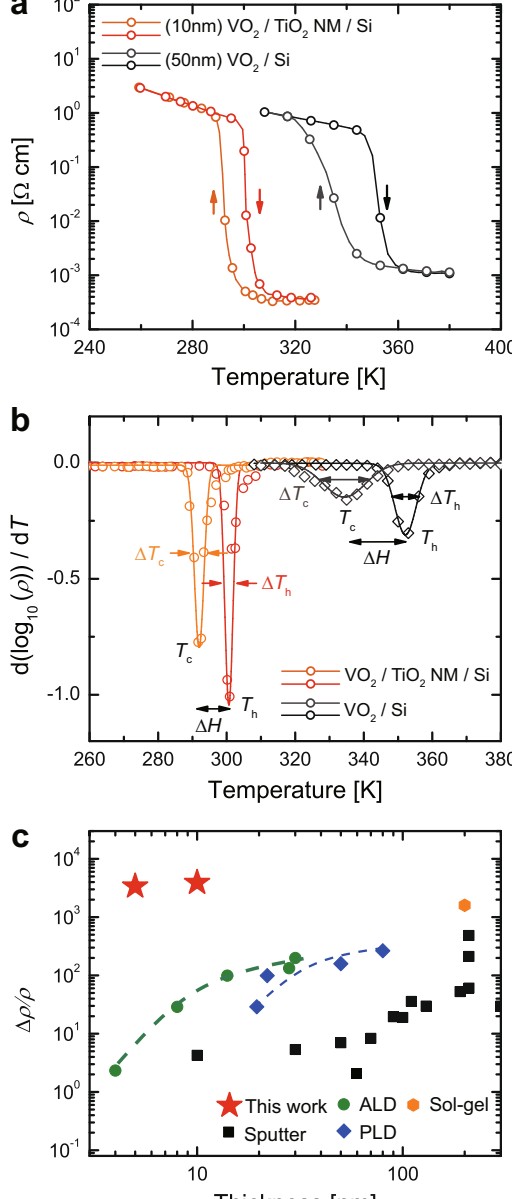

**Fig. 5 Single-crystalline VO$_2$ films with steep phase transition on TiO$_2$-NM-templated Si substrate. a** Temperature-dependent resistivity modulation near $T_{MIT}$ in 10-nm-thick VO$_2$ films on TiO$_2$ NM/SiO$_2$/Si (red lines) and 50-nm-thick VO$_2$ films on SiO$_2$/Si (black lines). **b** Comparison of various metal-insulator transition properties (d(log$_{10}$($\rho$))/d$T$, $T_h$, $T_c$, $\triangle H$, $\triangle T_h$, $\triangle T_c$) between 10-nm-thick VO$_2$ films on TiO$_2$ NM/SiO$_2$/Si (red lines) and 50-nm-thick VO$_2$ films on SiO$_2$/Si (black lines). Note that the transition width and sharpness, as well as the resistivity ratio, were substantially improved in VO$_2$ films with TiO$_2$ NM, compared to those without TiO$_2$ NM. **c** Benchmark of resistivity ratio $\Delta\rho/\rho = (\rho_{T_{MIT}-15K} - \rho_{T_{MIT}+15K})/\rho_{T_{MIT}+15K}$ for VO$_2$ films on Si substrates across the MIT. For a direct comparison, all films were grown on oxide-coated Si substrates using various growth techniques: (sputtering (black square), ALD (green circle), PLD (blue diamond), and sol-gel (orange hexagon)) reported in the previous literature (Supplementary Table 1). While deteriorated $\Delta\rho/\rho$ was observed in VO$_2$ films directly grown on Si due to the polycrystallinity of the films and the formation of a defective interfacial layer, epitaxial growth of VO$_2$ films (5 and 10 nm) guided by transferred TiO$_2$ NM enables integration of correlated oxides on silicon substrates with the highest modulation of resistivity ratio across the metal-insulator transition (red stars).

TiO$_2$ NM templates between VO$_2$ and Si enable the growth of epitaxial VO$_2$ films with crystallographic perfection free from defects.

To benchmark the quality of the VO$_2$ thin films on the TiO$_2$-NM-templated SiO$_2$/Si substrates in terms of MIT, temperature-dependent resistivity was measured for VO$_2$ films depending on the existence of transferred TiO$_2$ NM. Higher resistivity modulation ($\Delta\rho/\rho \sim 3.3 \times 10^3$) was observed across $T_{MIT} \sim 296$ K in both 5-nm-thick and 10-nm-thick VO$_2$ on TiO$_2$ NM/SiO$_2$/Si (red line in Fig. 5a and Supplementary Fig. 13b) than in 10-nm-thick VO$_2$ on SiO$_2$/Si (Supplementary Fig. 13a) and even in 50-nm-thick VO$_2$ on SiO$_2$/Si ($\Delta\rho/\rho \sim 5.2 \times 10^2$ across the $T_{MIT} \sim 343$ K) (black line in Fig. 5a). In addition to the resistivity ratio, other MIT properties (d(log$_{10}$($\rho$))/d$T$, $T_h$, $T_c$, $\triangle H$, $\triangle T_h$, $\triangle T_c$) were characterized as shown in Fig. 5b[32]. d(log$_{10}$($\rho$))/d$T$ was plotted and fitted with a Gaussian function. $T_h$ and $T_c$ were then defined at the peak position of the Gaussian and the transition width was calculated from the difference ($\triangle H = T_h - T_c$). The transition sharpness $\triangle T_h$ and $\triangle T_c$ was estimated from the full-width at half-maximum of the Gaussian peak. In fact, the transition width and sharpness, as well as

the resistivity ratio, were significantly improved in the VO$_2$ films with TiO$_2$ NM (i.e., 8.70 and 3.3 K for $\triangle H$ and $\triangle T_c$, respectively) compared to those without TiO$_2$ NM (i.e., 17.94 and 14.54 K for $\triangle H$ and $\triangle T_c$, respectively). The sharpened phase transition is attributed to the single-crystalline nature of the VO$_2$ films perfectly aligned with the underlying single-crystal TiO$_2$ NM templates. Moreover, the transferred single-crystal TiO$_2$ NM forms coherently tensile-strained VO$_2$ films; this epitaxial strain leads to a $T_{MI}$ shift close to room temperature in VO$_2$ films on TiO$_2$ NM/SiO$_2$/Si compared to relaxed VO$_2$ films on SiO$_2$/Si substrates[27,28,32]. Along with steep MIT under temperature, these single-crystalline VO$_2$ films on TiO$_2$ NM/Si show high endurance during thermal and electrical cycling. Both thermally induced MIT and electrically induced MIT ($I_{on}/I_{off} > 10^3$) were consistently observed without any drift during the multiple cycles of thermal switching (Supplementary Fig. 14) and electrical switching (Supplementary Fig. 15), respectively.

## Discussion
It should be noted that the integration of VO$_2$ films with excellent electrical properties on a Si substrate has proved challenging due to fundamental limitations. A representative set of data of MIT properties from VO$_2$ thin films on Si (or oxide-coated Si or buffered Si) substrates was compiled to allow for a direct comparison with those from our films (Fig. 5c, Supplementary Table 1, and Supplementary Fig. 16). Despite previous attempts to directly grow VO$_2$ thin films on Si substrates using various deposition techniques, $\triangle\rho/\rho$ across the $T_{MI}$ deteriorated due to the formation of polycrystallinity of the films. Notably, a further reduction in $\triangle\rho/\rho$ was found to occur with decreasing film thickness, likely due to the formation of defective interfacial layers (e.g., by thermodynamic reaction of VO$_2$ with Si (or SiO$_2$)); the cation-to-anion stoichiometry was not maintained in the VO$_2$ films near the interface[33,34]. If this compositional variation (i.e., off-stoichiometry) contains a significant fraction of the films, the resistivity ratio and transition sharpness will significantly degrade in the regime of an ultrathin thickness (<40 nm), as observed in our 10-nm-thick VO$_2$ films on SiO$_2$/Si and Al$_2$O$_3$-buffered Si (Supplementary Figs. 13, 16); steeper transition cannot be engineered simply by direct deposition of thin VO$_2$ films on Si substrates due to the existence of interfacial layers and the substantial density of defects.

However, the $TiO_2$ template NM on Si substrates permits epitaxy with $VO_2$ thin films due to its identical crystal structure ($P4_2/mnm$) and small lattice mismatch ($f_a \sim 0.86\%$). The transferred NM could play a crucial role as a seed layer for epitaxy, and artificially allows the first demonstration on heterogeneous integration of single-crystalline $VO_2$ on Si substrates, which are commonly employed in electronic devices. Furthermore, the $TiO_2$ NM is likely to prevent subsequent reaction between $VO_2$ and the Si substrate, and allows excellent control over the V-oxidation states without any extended defects[33,34]. Thus, epitaxial growth of ultrathin $VO_2$ films guided by transferred $TiO_2$ NM enables integration of correlated oxides with unprecedented modulation of the resistivity ratio ($\triangle\rho/\rho > 10^3$) across the MIT in the regime of an ultrathin thickness (5–10 nm) on Si substrates (see red stars in Fig. 5c).

In summary, heterogeneous integration of a freestanding single-crystalline rutile oxide NM was achieved by exploiting selective oxidation and dissolution of isostructural sacrificial layers. Despite a mild pH condition under a dilute $H_2O_2$ solution at room temperature, $VO_2$ sacrificial films are spontaneously oxidized by $H_2O_2$ and transformed to layered crystals, which in turn could be exfoliated and dispersed in solution to release and transfer the millimeter-scale $TiO_2$ NM with controlled thickness on oxide-coated Si substrates. Owing to the nearly perfect single-crystallinity of the transferred rutile $TiO_2$ NM, this lattice-matched single-crystal template permits heterogeneous integration of epitaxial $VO_2$ films on Si substrates; exceptional MIT characteristics in terms of transition sharpness ($\triangle T_h \sim 2.85$ K and $\triangle T_c \sim 3.3$ K) and resistivity ratio ($\triangle\rho/\rho > 10^3$) were realized in ultrathin $VO_2$ films integrated on Si substrates, benefitting from the superior quality of epitaxial oxide films.

Our strategy to release and transfer a freestanding epitaxial rutile oxide NM for a MIT will provide an unprecedented platform using emergent phenomena in epitaxial oxide heterostructures to be integrated with current state-of-the-art Si-based technology (e.g., integrated electronics[3,6] and photonics[13,35] on Si). For example, our epitaxial oxide NM with steep phase transition could deliver a hybrid optical modulator with a high extinction ratio, low loss, and high modulation speed integrated on Si waveguides[13,35,36]. Moreover, our approach for single-crystalline oxide NMs with high thermal stability are generally applicable for heterogeneous integration of high-quality oxide NM with any rutile crystal structures on Si substrates (e.g., epitaxial growth of metallic $RuO_2$ single-crystalline NMs on transferred $TiO_2$ NM/Si or direct transfer of metallic $RuO_2$ single-crystal NMs on Si, Supplementary Figs. 17–20); our study enables us to extend the materials spectrum for freestanding single-crystalline rutile oxide NM by developing a new combination of sacrificial layer and etchant. It also offers a unique opportunity for new types of artificial heterostructures by stacking multi-functional oxide NMs with other 2D/3D single-crystalline NMs (e.g., III–V, exfoliated 2D layered materials, complex oxides)[10,17] or by controlling twisted angle between two misaligned sheets of oxide NMs (similar to twisted bilayer graphene heterostructure with atomic and electronic reconstruction)[37,38] for novel interfacial physics and a new generation of emergent devices.

## Methods

**Epitaxial film growth**. The epitaxial $VO_2$ thin films (~17 nm) were grown on (001) $TiO_2$ single-crystal substrates, followed by the growth of $TiO_2$ films (10–70 nm) by PLD. First, (001) $TiO_2$ single-crystal substrates (Shinkosha CO., LTD) with lateral size of up to 5 mm × 5 mm were loaded into the PLD chamber, which was then evacuated to a base pressure of ~$1 \times 10^{-6}$ Torr. The rotating $V_2O_5$ and $TiO_2$ targets were then ablated by focusing a KrF excimer laser (Coherent Complex Pro 102 F, $\lambda = 248$ nm) with a fluence of 1 J/cm² and a repetition rate of 1 Hz. The growth of $VO_2$ films was performed at fixed $P_{O_2} = 15$ mTorr and $T_g = 300$ °C, which were selected to induce a steep MIT near room temperature from coherently tensile-strained $VO_2$ films. Subsequently, $TiO_2$ films were grown on $VO_2$ templates under $P_{O_2} = 28$ mTorr and $T_g = 300$ °C. After the growth of heterostructure, the samples were cooled to room temperature at a rate of 20 °C/min. The epitaxial $VO_2$ film (5–10 nm) was grown on the

transferred $TiO_2$ NM/Si with an identical growth condition of that of $VO_2$ sacrificial layers on the (001) $TiO_2$ substrate.

**Release and transfer of freestanding $TiO_2$ NMs**. To release the freestanding $TiO_2$ NM from the $VO_2/TiO_2$ substrates, the surface of the grown heterostructure was adhered onto a rigid supporting layer (e.g., polydimethylsiloxane, thermal release tape (Haeun Chemtec, RP70N5)). The structure was immersed in a dilute $H_2O_2$ solution until the sacrificial $VO_2$ layer was completely dissolved. To transfer the released $TiO_2$ NM to other substrates, $TiO_2$ NM was placed onto an oxide-coated silicon substrate and exposed to an appropriate temperature. Finally, the freestanding NM remained on the silicon substrate after peeling off the rigid supporting layer slowly. The structure was immersed in a dilute 10% $H_2O_2$ solution (i.e., 50 ml of 35% $H_2O_2$ + 150 ml of $H_2O$) at room temperature until the sacrificial $VO_2$ layer was completely dissolved, with the freestanding NM left on the rigid supporting layer. After dissolving the $VO_2$ layers in $H_2O_2$, $H_2O$ residue on the released $TiO_2$ surface was evaporated in a vacuum desiccator for 10 min. To transfer the released $TiO_2$ NM to other substrates (such as thermally grown $SiO_2$-coated silicon), the $TiO_2$ NM/rigid supporting layer was attached to a oxide-coated silicon substrate and was heated to 60–80 °C for 10 min. Finally, the freestanding NM remained on the silicon substrate after peeling off the rigid supporting layer slowly.

**SEM, EBSD, and AFM measurements**. OM images were recorded using BX53M (Olympus, Tokyo, Japan) microscope, equipped with an objective MPlanFL N (Olympus) and i-solution IMT cam CCD camera. The SEM and EBSD measurements were made using an XL30SFEG and FEI Helios Nanolab 650 equipped with an EBSD detector, respectively. The EBSD pattern was acquired by an EDAX Hikari EBSD camera while the sample was tilted at 70° and scanned with an electron beam of 25 nA at 20 kV. The measured data were analyzed using TSL OIM Analysis7. The AFM images were obtained using a VEECO Dimension 3100 in tapping mode.

**X-ray diffraction (XRD) and X-ray absorption spectroscopy (XAS)**. High-resolution x-ray scattering was performed by using synchrotron radiation at the 3D XRS ($\lambda \sim 0.12398$ nm, energy ~10 keV at Si (111)) beamline of Pohang Light Source-II (PLS-II, Pohang, Republic of Korea), and using an in-house HRXRD (Bruker Discover 8 X-ray diffractometer) with Cu $K_{\alpha1}$ radiation ($\lambda \sim 0.15406$ nm). The detailed information on in-plane and out-of-plane lattice parameters and strain states of each films and NM was obtained by using both symmetric 2θ-ω scan and asymmetric RSM around the (112) reflection. X-ray absorption spectroscopy (XAS) was performed using the 2 A MS beamline at PLS-II. The total electron yield mode with an energy resolution of ~0.1 eV was used for measurement at a base pressure of $5 \times 10^{-10}$ Torr in the analysis chamber by measuring the sample current ($I_1$) divided by the beam current ($I_0$) to remove the variation of the beam intensity.

**Scanning transmission electron microscope (STEM)**. Two sample types were prepared for plane-view STEM imaging of transferred $TiO_2$ NM on a carbon TEM grid and cross-sectional STEM imaging of transferred $TiO_2$ NM and $VO_2/TiO_2$ hetero-NM on $SiO_2$/Si. For the plane-view observation, the $TiO_2$ film was directly transferred onto the carbon TEM grid; $TiO_2/VO_2/TiO_2$ epitaxial heterostructure was physically attached to the PDMS with the carbon TEM grid and immersed in a dilute $H_2O_2$ solution. After selective oxidation and dissolution of the $VO_2$ sacrificial layer, the single-crystal $TiO_2$ NM was naturally released and attached to the carbon TEM grid (Supplementary Fig. 4). For the cross-sectional observation, the transferred $TiO_2$ NM and $VO_2/TiO_2$ hetero-NM on $SiO_2$/Si substrates were prepared by a focused ion beam (FIB) system (Helios G3, FEI), in which the samples were thinned by a Ga ion beam. The atomic structures were observed using a STEM (JEOL ARM 200 F, JEOL Ltd., Japan) with a fifth-order aberration corrector (ASCOR, CEOS GmbH, Heidelberg, Germany); the probe diameter and convergence angle of the beam were ~0.7 Å and ~27 mrad under an acceleration voltage of 200 kV, respectively. The collection semi-angles of the detectors for HAADF imaging were 54–210 mrad, and those for ABF imaging were 8–16 mrad to detect light elements (i.e., oxygen). The obtained STEM images were local difference filtered to reduce background noise (HREM Research Inc., Japan).

**Resistivity measurements**. The resistivity ($\rho$) was measured in the van der Pauw geometry during heating and cooling from 260 to 380 K by using a chamber probe station equipped with a Hall measurement system and a temperature control system. The resistivity change over the MIT ($\triangle\rho/\rho$) was defined as $\triangle\rho/\rho = (\rho_{T_{MIT}-15K} - \rho_{T_{MIT}+15K})/\rho_{T_{MIT}+15K}$. The fitting of the derivative of $\log_{10}(\rho)$ as a function of temperature (K) was performed based on the Gaussian function. $T_h$ and $T_c$ were then defined at the peak position of the Gaussian and the transition width was calculated from the difference ($\triangle H = T_h - T_c$). The transition sharpness $\triangle T_h$ and $\triangle T_c$ was estimated from the full-width at half-maximum of the Gaussian peak.

## Data availability

All relevant data within the article are available from the corresponding authors on reasonable request.

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

## Acknowledgements

We acknowledge support for this work by the Basic Science Research Program (2020R1A4A1018935 and 2020R1A2C2006389) through the National Research Foundation of Korea (NRF) funded by the Ministry of Science and ICT and the Korea Basic Science Institute (National Research Facilities and Equipment Center) Grant funded by the Ministry of Education (2020R1A6C101A202). D.K.L., Y.P., and J.S. acknowledge the support by the Samsung Research Funding & Incubation Center of Samsung Electronics under Project Number SRFC-TA1703-09. Y.P. acknowledges the support by the Basic Science Research Program through the National Research Foundation of Korea (NRF) funded by the Ministry of Education (2020R1A6A3A13075514). This study was partially supported by Brain Korea 21 Four for education and research center for future materials (F21YY7105002).

## Author contributions

J.S., D.K.L., and Y.P. conceived the idea and designed the study with assistance from C.-B.E.; D.K.L and Y.P. developed the fabrication process of freestanding nanomembranes, performed the growth of heterostructures, XRD, XAS, SEM, EBSD, AFM, and electrical transport measurement with assistance from J.P. and Y.K.; H.S., G.-Y.K., and S.-Y.C. characterized the nanomembranes by STEM; Y.K. assisted synchrotron spectroscopy measurement; J.S., D.K.L., Y.P., S.-Y.C., and H.S. wrote the manuscript and all authors commented on it; J.S. directed the overall research.

## Competing interests

The authors declare the following competing financial interest(s): D.K.L., Y.P., H.S., S.-Y.C., and J.S. are co-inventors on a patent application based on the results of this work filed by Pohang University of Science and Technology.
