## [Peer Review File · Nature Communications]

Heterogeneous integration of single-crystalline rutile nanomembranes with steep phase transition on silicon substratesEditorial Note: Parts of this Peer Review File have been redacted as indicated to maintain the confidentiality of unpublished data.

REVIEWER COMMENTS

Reviewer #1 (Remarks to the Author):

The authors report an etching-transfer-growth process for the development of heterogenous structure of VO₂/TiO₂/Si. The key step is the fast etching of interfacial VO₂ layer. The quality of the free standing TiO₂ membrane is astonishing. The VO₂ film on TiO₂/Si is of high quality as well, confirmed by the structural analysis and transport studies. In general, the method developed here is interesting and promising for the development of heterostructures. I would suggest publishing this manuscript after following minor questions are addressed:

1. Are there any residue of the rigid supporting layer (PDMS) left on TiO₂ NM?
2. In Supplementary Figure 1, the authors demonstrate that there are no VO₂ residue on TiO₂ substrate after H₂O₂ treatment. Have the authors characterized the VO₂ residue or any damages (e.g. roughened surface, defective boundaries) on the surface of TiO₂ NM which was in contact with the VO₂ sacrificial layer?
3. The reported method seems to be limited to VO₂/TiO₂ system which is different from either remote epitaxy or SAO sacrificial layer approach. How would the authors address this issue?
4. It seems that TiO₂ can not be removed if VO₂ is the target material. This would influence the design of device structures. For example, if an electrode/VO₂/Si heterostructure is needed, how would the authors develop it?

Reviewer #2 (Remarks to the Author):

The manuscript by D.K. Lee et al is concerned with making and measuring electrical conductivity and structural property of VO₂ layers grown on TiO₂ and making thin structures that can be integrated with silicon substrates. The material system is of current interest and potentially useful in different areas in optical and electronic applications. This work closely follows several papers on this topic (identical goals to make membranes for different device hetero-integration schemes on silicon etc and should be discussed as prior work in the manuscript) as noted below:

- Pellegrino, L, et al. "Multistate Memory Devices Based on Free-standing VO₂/TiO₂ Microstructures Driven by Joule Self-Heating." *Advanced Materials* 24.21 (2012): 2929-2934.
- Yamasaki, S, et al. "Metal-insulator transition in free-standing VO₂/TiO₂ microstructures through low-power Joule heating." *Applied Physics Express* 7.2 (2014): 023201.
- Sim, J. S., et al. "Suspended sub-50 nm vanadium dioxide membrane transistors: fabrication and ionic liquid gating studies." *Nanoscale* 4.22 (2012): 7056-7062.

The authors have examined etching approaches both similar and slightly different from the above manuscripts to make nanomembrane structures with TiO₂ and VO₂ on top. Given the extensive prior literature in this field, it will be important for the authors to go beyond simply measuring resistance versus temperature as a metric for VO₂ growth. The above papers from nearly a decade back all report sophisticated electrically-wired functional devices (eg ionic liquid FETs; multi-state memory etc).

The present manuscript falls short of this clearly, although the authors have indeed made a systematic effort to structurally characterize their films grown on TiO₂ by TEM methods and X-ray diffraction. But this aspect is not particularly new or original, as TiO₂ is historically one of the chosen substrates for VO₂ growth and hundreds of papers on structured/strain etc already exist on this same structure. There is also no particularly new physical property that is striking, despite observing a slightly more narrower IMT curve in thin films, but that is straightforward to engineer by synthesis

conditions as I am sure the authors are aware of (since this behavior is not intrinsic to VO₂ but due to VO₂ growth on TiO₂; and why is 10nm special, not clear to the reader?). There is also no thermal cycling or electrical fatigue data presented on the fabricated films and this is certainly essential to validate the process. Otherwise the resistance curve alone is not useful even if it seems sharp for the first switching measurement, since any drift in it will render the claimed applications irrelevant.

In this referee's opinion, original data to demonstrate some viable device or new concept with VO₂-TiO₂ that is uniquely enabled by some process feature reported in this study (going beyond the above references) will be desired for a major journal along with cycling data to show that the reported fabrication process is reliable and reproducible for making useful VO₂ devices.

Reviewer #3 (Remarks to the Author):

Lee et al. report on monocrystalline nanomembranes (NM) of TiO₂ and its transfer to silicon substrates for use as template for VO₂ heteroepitaxy. The VO₂ films show very sharp M-I transitions. The main novelty of the paper is the use of VO₂ sacrificial layers (SL), deposited epitaxially on TiO₂ substrates and removed by a H₂O₂ solution. Compared to the popular Sr₃Al₂O₆ (SAO), VO₂ SL presents a lower lattice mismatch with TiO₂, expanding the range of epitaxial oxides that can be prepared as NM by etching of oxide SLs. In addition, VO₂ is much stable to air exposure than SAO, thus overcoming an important practical limitation of SAO.

The results increase the current capabilities of oxide NMs and thus I consider the results to be relevant. However, before recommending publication in Nature Communications, the authors should review the paper considering several points.

1) In Introduction (p. 4, l. 75) the authors state "...the development of oxide NM has been limited for perovskite structure so far^{1,2,4,14,20,24}; to extend the materials spectrum for freestanding oxide NM, new combination of sacrificial layer and etchant needs to be developed..."

However, the recent paper cited in ref. 1 (Kum, H. S. et al. Heterogeneous integration of single-crystalline complex-oxide membranes. Nature 578, 75-81, (2020)) shows that oxide heteroepitaxy is possible when oxide substrates are covered by a few monolayers of graphene. In this reference, high quality oxide NM of perovskites, spinels and garnets are fabricated by this remote epitaxy method. Please revise the written sentence.

2) XRD scans shown in the manuscript were measured using synchrotron radiation. Similar scans would be useful, at least some of them presented as Supplementary Information, (for example to other authors who may try to replicate the fabrication method in the future).

3) The VO₂ SL is deposited at 300 °C. Please indicate, or confirm, if the deposition conditions for VO₂ on the transferred TiO₂ NM are the same.

4) The indicated area of the NM is "a few millimeters". Can be obtained and transferred NMs of larger area (at least the largest areas usual in standard PLD, around 10x10 or 10x8 mm²)? How does the etching time depend on the area?

5) Can the authors provide more information/results on the mechanical stability of the transferred TiO₂ NMs? The mechanical stability could be limited by stress at the VO₂/TiO₂ interface (f is around 0.86%) or at the TiO₂/Si interface (TEC mismatch).

5A) Are VO₂ films tens of nm thick on TiO₂/Si stable?

5B) Are films (VO₂ or other oxides) deposited at higher temperature (example 700 °C) stable?

6) Can another oxide, different from TiO₂, be deposited on VO₂ SL, or can it be deposited on TiO₂ / VO₂ before etching?

7) Can the authors illustrate the potential of the TiO₂ NM by depositing other functional oxide(s)?

8) The authors compare the properties of the VO₂ film deposited on the transferred NM to VO₂ films grown on SiO₂/Si. The films on SiO₂/Si are polycrystalline and thus the enhancement of properties is not surprising. Can it be compared to VO₂ films grown epitaxially directly on buffered Si(001)?

Detailed list of revisions to the manuscript

The following revisions have been made to the manuscript.

1. About the manuscript

We have revised the following sentences in the manuscript.

page 2, line 10	page 3, line 4 to line 8
page 3, line 17	page 6, line 9 to line 13
page 6, line 20 to line 21	page 8, line 8
page 11, line 1 to line 5	page 11, line 18 to line 20
page 12, line 1 to line 2	page 13, line 5 to line 12
page 14, line 3	page 14, line 11 to line 12

2. About the Supplementary Information.

We have revised the Supplementary Information as follows.

Supplementary Figs. 4, 5, 9, 10, 12, 14, 15, 16, 17, 18, 19, 20 have been added in Supplementary Information.

3. We added Mr. Jinheon Park, who contributed on the epitaxial growth of RuO₂ layers, as a co-authors of our revised manuscript.
4. We also fixed all editorial deficiencies for the publication of Nature Communications.

Response to the reviewer's comments (Reviewer #1)

➤ Comment 1

The authors report an etching-transfer-growth process for the development of heterogenous structure of VO₂/TiO₂/Si. The key step is the fast etching of interfacial VO₂ layer. The quality of the freestanding TiO₂ membrane is astonishing. The VO₂ film on TiO₂/Si is of high quality as well, confirmed by the structural analysis and transport studies. In general, the method developed here is interesting and promising for the development of heterostructures. I would suggest publishing this manuscript after following minor questions are addressed:

Responses: We thank the reviewer #1 for his/her compliment and very positive comments on our work. We are pleased to recognize his/her recommendation for publishing our work in Nature Communications. We appreciate the reviewer #1 for his/her valuable time and effort in reviewing our manuscript. The careful comments and questions were very helpful to improve our manuscript. We have comprehensively revised our manuscript to provide more experimental data based on his/her insightful comments. Our responses are presented point by point in the following passages.

➤ **Comment 2**

1. Are there any residue of the rigid supporting layer (PDMS) left on TiO₂ NM?

Response: We thank reviewer #1 for clarifying comments. As the reviewer #1 is concerned, there are residues (e.g., polymers or carbons) from the supporting layer (polydimethylsiloxane (PDMS) or thermal released tape (TRT)) on the surface of transferred TiO₂ NM (See **Fig. R1b**). However, we would like to point out that these polymer residues on TiO₂ NM was perfectly removed using our modified cleaning method based on previously reported RCA cleaning (see *ACS Nano*, **5**, 9144-9153, (2011)).

Below is our detailed procedure for the removal of residues on the surfaces of TiO₂ NM

Step 1: Transferred TiO₂ NM/Si was immersed in acetone (30 minutes), isopropyl alcohol (10 minutes), and de-ionized water (10 minutes) to remove organic residues.

Step 2: To strengthen the bonding between the TiO₂ NM and Si substrate and prevent delamination of the TiO₂ NM during subsequent cleaning process, transferred TiO₂ NM/Si was annealed at 500 °C for 3 hours.

Step 3: After thermal annealing, transferred TiO₂ NM/Si was immersed in the solution of 5:1:1 H₂O / NH₄OH / H₂O₂ at 80 °C for 10 minutes to removing residual organic contaminants.

Step 4: And then, residual metal contaminates was removed by using a solution of 5:1:1 H₂O / HCl / H₂O₂ at 80 °C for 10 minutes.

Step 5: Repeat ‘Step 3’ before growing VO₂ epitaxial film on TiO₂ NM.

To compare the surface quality of TiO₂ NM during transfer and subsequent cleaning process, we performed atomic force microscopy (AFM) of “as-grown” TiO₂/VO₂ epitaxial films on the TiO₂ single crystal substrates, “as-transferred” TiO₂ NM on silicon substrate, and TiO₂ NM on silicon substrate “after the cleaning process” (**Fig. R1**).

After transfer process by using PDMS or TRT, average roughness (R_a) increased from 0.089 nm (**Fig. R1a**) to 0.180 nm or 0.359 nm (**Fig. R1b**), respectively, due to the existence of

residues on the surface. Our cleaning methods revert R_a down to ~ 0.09 nm by effectively removing these residues (**Fig. R1c**); the surface of the TiO_2 NM on Si is as good as that of as-grown TiO_2 epitaxial films on VO_2/TiO_2 ; the surface quality is significantly improved by erasing these residues from rigid supporting layers (PDMS or TRT). We would like to emphasize that this cleaning method is critical for growing high-quality epitaxial VO_2 films on TiO_2 NM/Si substrates (**Fig. 4** in the manuscript).

Changes (Page 8 in the manuscript and SI): We added some sentences on cleaning procedure in the manuscript and AFM images during the cleaning procedure in the **Supplementary Figure 9** to share the critical process for epitaxial growth of VO_2 films on transferred TiO_2 NM.

Figure R1 Atomic force microscopy image of **a.** “as-grown” TiO_2/VO_2 epitaxial films on the TiO_2 single crystal substrates, **b.** “as-transferred” TiO_2 NM on silicon substrate, **c.** transferred TiO_2 NM on silicon substrate “after our RCA cleaning process”.

➤ **Comment 3**

2. In Supplementary Figure 1, the authors demonstrate that there are no VO₂ residue on TiO₂ substrate after H₂O₂ treatment. Have the authors characterized the VO₂ residue or any damages (e.g. roughened surface, defective boundaries) on the surface of TiO₂ NM which was in contact with the VO₂ sacrificial layer?

Response: We appreciate the reviewer #1 for the clarifying comments. As the reviewer #1 commented, we already characterized the surface quality of etched TiO₂ NM, which was in contact with the VO₂ sacrificial layer, by AFM prior to transferring on the Si substrates (i.e., released TiO₂ NM on the supporting layer). As shown in **Fig. R2**, except height modulation by the slight bending due to the elastic supporting layer, no VO₂ residue or any damages was observed at the surface of TiO₂ NM after H₂O₂ treatment.

Changes (SI): We added the AFM image and height profile of released TiO₂ NM on supporting layer in **Supplementary Figure 10**.

Figure R2 a. Atomic force microscopy image and **b.** depth profile along the x-axis direction of released TiO₂ NM on supporting layer.

➤ **Comment 4**

3. The reported method seems to be limited to VO₂/TiO₂ system which is different from either remote epitaxy or SAO sacrificial layer approach. How would the authors address this issue?

Response: We appreciate the reviewer #1 for very constructive comment. The reviewer #1 claimed that our work may be limited to only VO₂/TiO₂ system. However, we would like to emphasize our work is generally applicable for the fabrication of “single-crystalline” rutile oxide nanomembranes (NM).

Figure R3 Heterogeneous integration of “single-crystalline” RuO₂ films on TiO₂ NM/Si. **a**, schematic of an epitaxial TiO₂/VO₂ heterostructure on TiO₂ mother substrate. **b**, The VO₂ layer is dissolved in H₂O₂ to release the top TiO₂ film with the mechanical supporting layer (e.g., PDMS and TRT). **c**, The freestanding TiO₂ NM is transferred onto the desired substrates (e.g., silicon). **d**, By removing the rigid supporting layer, single-crystalline rutile oxide NM is heterogeneously integrated into a silicon substrate. **e**, epitaxial RuO₂ film is grown on the TiO₂-NM-templated Si substrates.

To generally demonstrate the heterogeneous integration of other rutile oxide single-crystalline layers on Si, we grew epitaxial RuO₂ films on TiO₂ NM/Si (Fig. R3). After the cleaning process to remove the residues on transferred TiO₂ NM, 9-nm-thick RuO₂ thin films, instead

of VO₂, were grown on TiO₂ NM/Si to realize RuO₂/TiO₂ epitaxial heterostructure integrated on Si substrates by pulsed laser deposition (**Fig. R3e**). As observed in symmetrical XRD 2θ-ω scans (**Fig. R4**), the intense (002) RuO₂ peak appeared at ~ 2θ = 47.8°, along with peaks related to the TiO₂ template (~ 2θ = 49.5°) and Si substrates (~ 2θ = 54.3°) (**Fig. R4b**). The location of (002) RuO₂ peak on TiO₂ NM/Si (above figure in **Fig. R4b**) was almost identical that of epitaxial RuO₂ films grown on (001) TiO₂ substrates (above figure in **Fig. R4d**); this result implicates that TiO₂ “single-crystalline” NM templates facilitate the formation of epitaxial RuO₂ films on Si substrates as well.

Figure R4 XRD of “single-crystalline” RuO₂ films on TiO₂ NM/Si. Schematic of **a**. RuO₂ epitaxial films grown on the TiO₂ NM / Si substrate and **c**. RuO₂ films grown on the (001) TiO₂ substrate. Corresponding symmetric 2θ-ω scans of **b**. RuO₂ / TiO₂ NM / Si and **d**. RuO₂ / (001) TiO₂ substrate. The location of (002) RuO₂ peak on TiO₂ NM/Si was almost identical that of epitaxial RuO₂ films grown on (001) TiO₂ substrates.

The cross sectional HAADF-STEM images confirm that the epitaxial growth of 9-nm-thick RuO₂ on TiO₂ NM can realize heterogeneous integration of single-crystal RuO₂ films on a Si substrates. Unlike VO₂ films, the RuO₂ film appears much brighter than TiO₂ NM because the atomic number of Ru ($Z_{\text{Ru}} = 44$) is much larger than that of Ti ($Z_{\text{Ti}} = 22$). The low magnification image reveals that the RuO₂ thin films were uniformly grown on TiO₂ NM with ~ 9 nm thickness (**Fig. R5 a**). The atomic-scale resolution image near the interface between RuO₂/TiO₂, indicated by yellow square in **Fig. R5 a**, is shown in **Fig. R5 b** (zone axis : [100] in both TiO₂ and RuO₂). The atomic columns of RuO₂ and TiO₂ are coherently matched at the interface with perfect rutile crystal without any noticeable defects; it means TiO₂ NM on Si allows the epitaxial growth of RuO₂ films; high-magnification HAADF-STEM image shows perfect registry of Ru atoms in the rutile structure and sharp interface from the contrast difference between RuO₂ and TiO₂ (**Fig. R5 b**). Selective area electron diffraction (SADP) spots on RuO₂/TiO₂ NM regions show that both RuO₂ film and TiO₂ NM have high quality of rutile crystal structures (**Fig. R5 c**). The out-of-plane direction diffraction spots (i.e., 002, indicated by blue square) are slightly separated due to lattice mismatch between RuO₂ and TiO₂, whereas the in-plane direction diffraction spots (i.e., 040, indicated by red square) are completely overlapped as a single spot, also confirming that epitaxy growth of RuO₂ thin films on TiO₂ NM. When comparing the lattice parameter $c/a(=c/b)$ ratio calculated from the SADP images with the c/a ratio in the bulk states, there were differences by - 0.98% and - 4.48% in TiO₂ and RuO₂, respectively. The large decrease of c/a in RuO₂ indicates that the RuO₂ films are fully strained by biaxial tensile strain along the in-plane direction; SADP results confirmed perfect epitaxy between TiO₂ NM and RuO₂ layers transferred on Si, and thus TiO₂ NM templates between RuO₂ and Si enable epitaxial growth of RuO₂ films, as well as VO₂ films, with crystallographic perfection free from defects.

Figure R5 STEM analysis of RuO₂/TiO₂ NM on SiO₂/Si substrate. **a**, The cross sectional HAADF-STEM image of the heterogeneous RuO₂/TiO₂ NM on SiO₂/Si. The RuO₂ film appears much brighter than TiO₂ NM because the atomic number of Ru ($Z_{\text{Ru}}=44$) is much larger than that of Ti ($Z_{\text{Ti}}=22$). The low magnification image reveals that the RuO₂ thin films were uniformly deposited on TiO₂ NM with 8.5 nm thickness. The atomic-scale resolution image near the interface between RuO₂/TiO₂, indicated by yellow square in **a**, is shown in **b** (zone axis : [100] in both TiO₂ and RuO₂). The atomic columns of RuO₂ and TiO₂ are coherently matched at the interface with perfect rutile crystal structure; it means TiO₂ NM on SiO₂/Si allows the epitaxy growth of RuO₂ thin films despite the large lattice mismatch ($\sim 2\%$). **c**. SADP images on RuO₂/TiO₂ NM regions. The sharp diffraction spots shows both RuO₂ film and TiO₂ NM have high quality of rutile crystal structures. The out-of-plane direction diffraction spots (i.e., 002, indicated by blue square) are slightly separated due to lattice mismatch between RuO₂ and TiO₂, whereas the in-plane direction diffraction spots (i.e., 040, indicated by red square) are completely overlapped as a single spot, also confirming that epitaxy growth of RuO₂ thin films on TiO₂ NM. When comparing the lattice parameter $c/a(=c/b)$ ratio calculated from the SADP images with the c/a ratio in the bulk states, there were differences by -0.98% and -4.48% in TiO₂ and RuO₂, respectively. The large decrease of c/a in RuO₂ indicates that the RuO₂ films are fully strained by biaxial tensile strain along the in-plane direction.

Our further experiments during the revision clearly demonstrates the “general” aspect of our approach for the production of single-crystalline NM with rutile oxide crystal system. TiO₂ NM may facilitate synthesis of other single-crystalline rutile oxide films such as RuO₂, SnO₂, NbO₂ and IrO₂ without any extended defects on the Si substrates and devices

As reviewer #1 mentioned, SAO sacrificial layer and remote epitaxy has been suggested for general synthetic methods for single-crystalline NM. First of all, a perovskite oxide membrane was gently released by dissolving water-soluble Sr₃Al₂O₆ sacrificial layers, but the moisture-sensitive nature of these layers prevents long-time exposure of the sacrificial layers, which

restricts practical application for heterogeneous integration of oxide NM, as reviewer #3 commented. Moreover, the development of oxide NM has been limited for perovskite structure (e.g., SrTiO₃, BaTiO₃ etc.). Therefore, our study enables to “extend” the materials spectrum for freestanding single-crystalline rutile oxide NM by developing new combination of VO₂ sacrificial layer and H₂O₂ etchant.

In addition, we would like to emphasize that remote epitaxy approach for oxide NM may not be suitable for synthesizing ultrathin oxide NM. To prevent oxidation of interfacial graphene to production of oxides membranes, epitaxial oxide films should be grown under vacuum ($< 5 \times 10^{-6}$ torr) at initial growth (about 5 – 10 nm) (*Nature* **578**, 75-81 (2020)). Oxide film growth under vacuum may generate oxygen vacancies in the oxide films and/or interrupt accurate control of stoichiometric ratio (e.g., the formation of oxygen deficient sub-oxide); the generation of defects in ultrathin regime near the interface could suppress the intrinsic properties of multi-functional oxide films. Moreover, the coalescence of localized nuclei in oxide layers on the graphene, along with restriction of oxygen environment during growth, prevents the layer-by-layer growth of VO₂ NM with atomic precision and high quality using remote epitaxy.

Recently, remote epitaxy of VO₂ NM was demonstrated by using graphene interfacial layer (*Nano Lett.* **20**, 33-42 (2020)), but VO₂ NM exhibited domain boundaries by rotational symmetry mismatch between monoclinic (100)_R-VO₂ films with a 2/m point group and rhombohedral (0001) Al₂O₃ substrates with a $\bar{3}m$ point group. The existence of these extended defects prohibits steep metal-insulator transition of VO₂ NM by remote epitaxy. In these respects, our approach for single-crystalline rutile oxide NM are “unique” and “generally applicable” for heterogeneous integration of oxide NM with any rutile crystal structures on Si substrates.

Changes (Page 4 and Page 14 in the manuscript and SI): We hope we fully addressed reviewer #1’s constructive comments and we also added a sentence on originality and generality of our fabrication method for “single-crystalline” rutile NM and the figures to Supplementary Figures 17, 18.

➤ **Comment 5**

4. It seems that TiO₂ cannot be removed if VO₂ is the target material. This would influence the design of device structures. For example, if an electrode/VO₂/Si heterostructure is needed, how would the authors develop it?

Response: We agree with the reviewer #1's remark. As the reviewer #1 commented, freestanding TiO₂ single-crystal NM is essentially required to grow epitaxial VO₂ film on the Si substrate, because transferred TiO₂ NM serves as an epitaxial template for heterogeneous integration of VO₂ on Si. However, due to layer-by-layer growth of epitaxial TiO₂ films, the thickness of TiO₂ template could be minimized down to a few nanometers between VO₂ and Si substrates. In terms of processing, this TiO₂ NM is likely to prevent chemical reaction between VO₂ and the Si substrates, and allows excellent control over the V-oxidation states without any defects; we would admit that TiO₂ NM is indispensable ingredient for the heterogeneous integration of single-crystalline VO₂ layers on Si substrates, but the its thickness could be minimized as thin as possible.

More generally, we can absolutely growth other "epitaxial" oxides with rutile structures, and this oxide layers can be released from TiO₂ mother substrates and transferred on Si substrates, as long as the oxide are insoluble under dilute H₂O₂ solution; Among many candidates of rutile oxides, RuO₂ was selected to prove the generality of our approach for "single-crystalline" rutile NM; other single-crystalline rutile oxide NMs (e.g., RuO₂ or IrO₂ NM) as a conducting electrodes, instead of TiO₂ NM, could be transferred on Si substrates; VO₂/electrode (i.e., RuO₂)/Si heterostructures could be realized using the same technique.

For RuO₂ electrode NM, we firstly grew 9 nm-thick RuO₂ films on the VO₂/TiO₂ heterostructure under $pO_2 = 40$ mTorr and $T_g = 400$ °C (**Fig. R6 a**). Then, we performed symmetric XRD 2θ - ω scans of RuO₂/VO₂ on TiO₂ substrates (**Fig. R6 e**). As expected, strong (002) RuO₂ peak appeared at $\sim 47.8^\circ$ in the sample. Then, rigid supporting layer was coated on the RuO₂/VO₂/TiO₂ heterostructure and immersed it into H₂O₂ solution at room temperature. RuO₂ NMs were released from the TiO₂ substrates by using selective etching of

VO₂ sacrificial layer (**Fig. R6 b**). After fully dissolving VO₂ sacrificial template, RuO₂ NMs were released, and then transferred on Si substrate (**Fig. R6 c**), followed by the removal of rigid supporting layer (**Fig. R6 d**). Symmetric 2 θ - ω scan by using synchrotron XRD showed that (002) RuO₂ peak appeared with (400) Si substrate (**Fig. R6 f**), which confirms freestanding epitaxial RuO₂ film was successfully transferred on the Si substrate.

Figure R6 Heterogeneous integration of RuO₂ NM on Si **a.** schematic of an epitaxial RuO₂/VO₂ heterostructure on TiO₂ mother substrate. **b,** The VO₂ layer is dissolved in H₂O₂ to release the top RuO₂ film with the mechanical supporting layer (e.g., PDMS and TRT). **c,** The freestanding RuO₂ NM is transferred onto the Si substrate **d,** By removing the rigid supporting layer, single-crystalline RuO₂ is heterogeneously integrated into a silicon substrate. Corresponding symmetric 2 θ - ω XRD scans of the **e.** Fig. R6 a and **f.** Fig. R6 d. The epitaxial RuO₂ film was successfully transferred on the Si substrate.

Thus, our strategy to release and transfer a freestanding epitaxial rutile oxide is quite versatile techniques and can be applicable to fabricate a number of combination of heterostructures; this will provide an unprecedented platform for emergent phenomena in various oxide heterostructures with various conductivity (i.e., conductors (e.g., RuO₂), semiconductors (e.g., SnO₂), insulators (e.g., TiO₂) and even emerging superconductivity (e.g., strained RuO₂)) to be integrated with mature Si-based heterostructures and devices.

Changes (Page 14 in the manuscript and SI): We hope we fully addressed reviewer #1's constructive comments and we also added a sentence on generality of our fabrication method for "single-crystalline" rutile NM and the figures to **Supplementary Figure 20**.

Response to the reviewer's comments (Reviewer #2)

➤ Comment 1

The manuscript by D. K. Lee et al is concerned with making and measuring electrical conductivity and structural property of VO₂ layers grown on TiO₂ and making thin structures that can be integrated with silicon substrates. The material system is of current interest and potentially useful in different areas in optical and electronic applications. This work closely follows several papers on this topic (identical goals to make membranes for different device hetero-integration schemes on silicon etc and should be discussed as prior work in the manuscript) as noted below:

- Pellegrino, L, et al. "Multistate Memory Devices Based on Free-standing VO₂/TiO₂ Microstructures Driven by Joule Self-Heating." *Advanced Materials* 24.21 (2012): 2929-2934.
- Yamasaki, S, et al. "Metal–insulator transition in free-standing VO₂/TiO₂ microstructures through low-power Joule heating." *Applied Physics Express* 7.2 (2014): 023201.
- Sim, J. S., et al. "Suspended sub-50 nm vanadium dioxide membrane transistors: fabrication and ionic liquid gating studies." *Nanoscale* 4.22 (2012): 7056-7062.

The authors have examined etching approaches both similar and slightly different from the above manuscripts to make nanomembrane structures with TiO₂ and VO₂ on top. Given the extensive prior literature in this field, it will be important for the authors to go beyond simply measuring resistance versus temperature as a metric for VO₂ growth. The above papers from nearly a decade back all report sophisticated electrically-wired functional devices (e.g. ionic liquid FETs; multi-state memory etc).

Response: First of all, we thank the reviewer #2 for his/her interest on our manuscript. We appreciate the reviewer #2 for his/her valuable time and effort in reviewing our manuscript. Despite the reviewer #2's interest on our study, he/she seems to doubt the originality of our work based on prior literatures on rutile nanomembranes. However, I would like to emphasize that our VO₂/TiO₂ nanomembranes are perfect "single-crystalline" without any extended defects. Our work clearly shows the first demonstration of single-crystalline VO₂/TiO₂

heterostructures on Si substrates; exceptional MIT characteristics in terms of transition sharpness ($\Delta T_h \sim 2.85$ K, $\Delta T_c \sim 3.3$ K) and resistivity ratio ($\Delta\rho/\rho > 10^3$) were realized in ultrathin VO₂ films integrated on Si substrates, benefitting from the superior quality of epitaxial oxide films.

We address reviewer #2's misunderstanding on our work and, more importantly, emphasize the scientific and technical significance of our study in the following responses, which we failed to deliver to reviewer #2 in our manuscript before revision.

First of all, in this comment, the reviewer #2 claimed that our work followed several literatures with identical goals to make membranes. **However, we would like to emphasize that previous literatures that the reviewer #2 referred to did not demonstrate perfect “single-crystalline” VO₂/TiO₂ hetero-membranes. These membranes showed relatively poor crystal quality and properties compared to our nanomembranes.**

For example, in the referred literatures by reviewer #2 (*Adv. Mater.* 24, 2929-2934 (2012), *Appl. Phys. Express* 7, 023201 (2014)), VO₂/TiO₂ hetero-structure were grown on the MgO substrates by domain matching with the following crystallographic relationship (TiO₂ (110)[001] \parallel MgO (100)[011]): Tetragonal rutile TiO₂ has lattice constants of $a = 4.5936$ Å and $c = 2.9586$ Å. MgO is NaCl type cubic with lattice constant of $a = 4.2123$ Å. Rutile (110) plane of TiO₂ exhibits a unit area of 2.959 Å \times 6.496 Å ($\sqrt{2}a$); [001] direction distance of TiO₂ ($c = 2.959$ Å) is close to the distance between same atoms in MgO (100) plane (i.e., $a/\sqrt{2} = 2.979$ Å) with lattice mismatch of - 0.67 % (*Jpn. J. Appl. Phys.* **44**, 3192-3195 (2005)). Despite preferential orientation of TiO₂ (or VO₂) films on MgO, however, twin boundary (or crystal grains) could be formed due to two possible orientations of TiO₂ (110)[001] \parallel MgO (100)[011] and TiO₂ (110)[001] \parallel MgO (100)[01 $\bar{1}$]. Moreover, 50 nm-thick interfacial MgTiO₃ layers could be formed in subsequent annealing by the chemical reaction between MgO and TiO₂ (*Jpn. J. Appl. Phys.* **44**, 3192-3195 (2005)). Thus, rutile TiO₂ film grown on the MgO substrate contained many defects such as stacking faults and dislocation (*Jpn. J. Appl. Phys.* **44**, 3192-3195 (2005)). Due to relatively poor crystal quality of VO₂/TiO₂ hetero structure on MgO substrate, full width at half maximum (FWHM) of rocking curves around (110) TiO₂ reflection ($\sim 3^\circ$, *Adv. Mater.* 24, 2929-2934 (2012)) from TiO₂ on MgO is

significantly greater than that from our TiO₂ nanomembranes (~ 0.07°) (Fig. R7 a). Moreover, much rougher surface was reported in VO₂/TiO₂ on MgO (R_{rms} ~ 1 nm) than that in our VO₂/TiO₂ NM/Si (R_a ~ 0.09 nm, R_{rms} ~ 0.14 nm) (Fig. R7 b).

As a result, poor structural quality with a number of defects in TiO₂ templates could degrade the sharpness and resistivity modulation of metal-insulator transition in VO₂ films on TiO₂/MgO (e.g., $\Delta\rho/\rho \sim 10^2$, $\Delta T_c \sim 12.5$ K with VO₂ thickness of 70 nm in Fig. 2 from *Adv. Mater.* 24, 2929-2934 (2012)) compared to our fully epitaxial VO₂ films on transferred TiO₂ NM/Si (i.e., $\Delta\rho/\rho > 10^3$, $\Delta T_c \sim 3.3$ K with VO₂ thickness of 10 nm). Likewise, VO₂ membranes grown on the Si₃N₄ substrate, which is referred from reviewer #2, showed polycrystalline nature of structural quality, which significantly suppressed metal-insulator transition (*Nanoscale* 4, 7056-7062 (2012), *Appl. Phys. Lett.* 87, 051910 (2005)); in terms of structural quality, VO₂ films on the Si₃N₄ are similar to directly grown VO₂ films on Si substrates (control sample in black line of Fig. 5a), which is completely different from fully epitaxial VO₂ films on TiO₂ NM/Si substrates.

Figure R7 a. Rocking curve of (002) VO₂ peak and (002) TiO₂ peak in the VO₂/TiO₂ NM on the Si substrate. FWHM is approximately 0.07°, which indicates all oxide films consisted of VO₂ and TiO₂ NM have high crystal quality. **b.** AFM image of VO₂/TiO₂ NM verifies very clean surface.

Again, our strategy to release and transfer TiO₂ single-crystalline NM enables the integration of nearly perfect single-crystalline VO₂ films with controlled thickness on Si substrates, which is never reported before. VO₂ film grown on the transferred TiO₂ NM had no distinct defects. And low-magnification ABF-STEM image shows that the VO₂/TiO₂ NM is free of defects even after transfer process from the wide field of view (**Fig. R8 a**). As a further evidence for exceptional crystalline quality, we scrutinized VO₂/TiO₂ NM integrated on Si using scanning transmission electron microscope (STEM). The high magnification ABF-STEM image sufficiently verifies the perfect registry of Ti (blue) and oxygen (red) atoms in the rutile structure (**Fig. R8 b**) in VO₂/TiO₂ NM. Furthermore, high magnification HAADF-STEM images with the [100] zone axis visualized the structural coherency of the freestanding oxide NM and VO₂ film (**Fig. R8 c**); the atomic resolution image implicates that the VO₂ layer is tightly constrained from the underlying TiO₂ NM, and thus TiO₂ NM templates between VO₂ and Si enable growth of epitaxial VO₂ films with crystallographic perfection free from defects such as stacking faults and dislocation, which enables steep phase transition at the ultra-thin VO₂ film.(i.e., under 10 nm).

Based on these multiple supporting data and arguments on the uniqueness and structural integrity of our “single-crystalline” NMs compared to previously reported membranes, we strongly opposed to reviewer #2’s under-appreciation of our work.

Changes (Page 3 in the manuscript and SI): Anyway, we added the prior literatures suggested by reviewer #2 in the reference section of our manuscript to emphasize the uniqueness of our work. Moreover, we added some sentences to emphasize the exceptional quality of our NMs and substantial figures on Rocking Curves and AFM on TiO₂/VO₂ NM on Si in **Supplementary Figure 12**.

Figure R8 a. Low magnification cross-sectional ABF-STEM images of the heterogeneous VO_2/TiO_2 NM on SiO_2/Si . The atomic scale ABF and HAADF-STEM image at the area indicated by the **b.** blue square and **c.** red square in **a** (zone axis : $[100]$ in TiO_2). ABF-STEM image sufficiently verifies the perfect registry of Ti (blue) and oxygen (red) atoms in the rutile structure. And, the VO_2 layer is tightly constrained from the underlying TiO_2 NM as shown in HAADF-STEM image

➤ **Comment 2**

The present manuscript falls short of this clearly, although the authors have indeed made a systematic effort to structurally characterize their films grown on TiO₂ by TEM methods and X-ray diffraction. But this aspect is not particularly new or original, as TiO₂ is historically one of the chosen substrates for VO₂ growth and hundreds of papers on structured/strain etc already exist on this same structure. There is also no particularly new physical property that is striking, despite observing a slightly more narrower IMT curve in thin films, but that is straightforward to engineer by synthesis conditions as I am sure the authors are aware of (since this behavior is not intrinsic to VO₂ but due to VO₂ growth on TiO₂; and why is 10 nm special, not clear to the reader?). There is also no thermal cycling or electrical fatigue data presented on the fabricated films and this is certainly essential to validate the process. Otherwise the resistance curve alone is not useful even if it seems sharp for the first switching measurement, since any drift in it will render the claimed applications irrelevant.

In this referee's opinion, original data to demonstrate some viable device or new concept with VO₂-TiO₂ that is uniquely enabled by some process feature reported in this study (going beyond the above references) will be desired for a major journal along with cycling data to show that the reported fabrication process is reliable and reproducible for making useful VO₂ devices

Responses: We strongly disagree with the reviewer #2's remark. Our work focus on the heterogeneous integration of "single-crystalline" functional oxide with steep metal-insulator transition on the Si substrates. Our work is different from previous VO₂/TiO₂ membranes with numerous defects in the literatures that reviewer #2 mentioned. Although thick TiO₂ substrates has been a well-known substrates for VO₂ growth, there has been no report on the "epitaxial" growth of VO₂ films on freestanding "nanometer-thick" "single-crystal" TiO₂ membranes, which our work demonstrated in the manuscript; our new strategy to release TiO₂ NM enables the first demonstration of perfectly "epitaxial" VO₂ films integrated on Si substrates. **This steeper and more enhanced MIT properties cannot be engineered simply by direct deposition of thin VO₂ films on Si or MgO substrates due to the existence of interfacial layers and the substantial density of defects as we already**

mentioned in our previous response.

Despite numerous effort to integrate “single-crystalline” oxide nanomembranes on silicon, heteroepitaxial growth of oxide films drastically limits the possible materials combinations due to the requirement of lattice matching between epilayer and substrates. In particular, epitaxial growth of VO₂ on Si is fundamentally impossible because direct growth of VO₂ on Si forms defective and polycrystalline layers. Despite significant efforts to improve structural and electrical characteristics of VO₂ thin films using the above buffer layers on Si, we would like to emphasize that high resistivity modulation ($\Delta\rho/\rho$) is difficult to be achieved in ultrathin regime of VO₂ thickness due to a number of interfacial defects and domain boundaries between buffer layer and VO₂ film (see Comment 10 from reviewer #3). Our work clearly demonstrates the heterogeneous integration of “single-crystalline” oxide films with steep metal-insulator transition on Si; this work will offer unique way to integrate the emergent phenomena of epitaxial oxide films with mature electronic and photonic devices.

Previously, for “single-crystalline” nanomembranes, SAO sacrificial layer and remote epitaxy has been suggested for general synthetic methods for “single-crystalline” NM (*Nat. Mater.* **15**, 1255-1260 (2016)). First of all, a perovskite oxide membrane was gently released by dissolving water-soluble Sr₃Al₂O₆ sacrificial layers, but the moisture-sensitive nature of these layers prevents long-time exposure of the sacrificial layers, which restricts practical application for heterogeneous integration of oxide NM. Moreover, the development of oxide NM has been limited for perovskite structure (e.g., SrTiO₃, BaTiO₃ etc.). Therefore, our study enables to extend the materials spectrum for freestanding single-crystalline rutile oxide NM by developing new combination of VO₂ sacrificial layer and H₂O₂ etchant.

In addition, remote epitaxy of VO₂ NM was recently demonstrated by using graphene interfacial layer (*Nano Lett.* **20**, 33-42 (2020)), but VO₂ NM exhibited domain boundaries by rotational symmetry mismatch between monoclinic (100)_R-VO₂ films with a 2/m point group and rhombohedral (0001) Al₂O₃ substrates with a $\bar{3}m$ point group. The existence of these extended defects prohibits steep metal-insulator transition of VO₂ NM by remote epitaxy. Furthermore, the coalescence of localized nuclei in oxide layers on the graphene and restriction of oxygen environment during growth prevents the layer-by-layer growth of VO₂

NM with atomic precision and high quality using remote epitaxy (*Nature* **578**, 75-81 (2020)). In these respects, our approach for single-crystalline rutile oxide NM are unique and generally applicable for heterogeneous integration of oxide NM with any rutile crystal structures on Si substrates.

Despite our strong disagreement of reviewer #2's claim on the originality of our work, we agree with the reviewer #2's suggestion on thermal and electrical cycling of "single-crystalline" VO₂ films on TiO₂ NM/Si, which is certainly essential to validate the process for the possible application of integrated VO₂ films on Si. First of all, we performed the repeated thermal cycling by measuring temperature-dependent sheet resistance at several times (270 K ~ 320 K) as shown in **Fig. R9 a**. Obviously, steep sheet resistance modulation with metal-insulator transition across $T_{MI} \sim 296$ K ($\Delta\rho/\rho \sim 10^3$) were observed in 10 nm-thick VO₂ on TiO₂ NM/Si more than 100 cycles (**Fig. R9 b**). No drift was observed during the multiple cycle of thermal switching of metal-insulator transition in VO₂ on TiO₂ NM/Si

Figure R9 a. Repeated measurement of temperature-dependent resistance measurement from 270 K to 320 K during 5 cycles. **b.** thermal cycling at 278 K and 308 K until 100 cycles in the 10 nm-thick VO₂ / 70 nm-thick TiO₂ NM.

In addition to thermal cycling, we also characterized threshold switching (i.e., voltage-triggered insulator-to-metal transition) after the fabrication of two-terminal devices with 5 μm

of lateral dimension between two Au/Ti electrodes (**Fig. R10 a**). Uniform and abrupt voltage-triggered threshold switching (i.e., sudden increase of current at threshold voltage) was observed in epitaxial VO₂ films integrated on Si substrates (**Fig. R10 b**); this observation indicates that reversible electric-field-induced switching from insulating phase to metallic phase occurs at consistent threshold voltage ($V_{th} \sim 5.7$ V) in the forward direction and subsequent metal-to-insulator transition at hold voltage ($V_h \sim 0.5$ V) in the reverse direction of sweep. This two terminal devices with “single-crystalline” showed high reliable selectivity of on and off states more than 100 cycles, which shows excellent the endurance of I_{on}/I_{off} ratio, V_{th} and V_h during DC bias sweep (Fig. R10 c, d), which can be potentially applicable for selectors and steep-slope switches in the future. Again, No drift was observed during the multiple cycle of electrical switching of metal-insulator transition in VO₂ on TiO₂ NM/Si

Although we did not focus on the device performance as an originality of our manuscript, we sufficiently proves that our single crystal VO₂ film integrated on the Si substrate was high endurance during thermal and electrical cycling as the reviewer #2 suggested.

Figure R10 a. Plane view of two-terminal threshold switch with single crystalline VO_2/TiO_2 NM on Si obtained by optical microscopy. **b.** the I-V characteristic of two-terminal single crystal VO_2 device consisted of Au/Ti electrodes on the VO_2/TiO_2 NM with 5 μm channel length. **c.** the DC endurance of the single crystal VO_2 device, which showed stable resistivity change. **d.** homogeneous V_{th} and V_h changes during DC bias endurance more than 100 cycles.

Changes (Page 12, SI) : We added some sentences in the manuscript and figures in the **Supplementary Figures 14 and 15** on the high reproducibility of metal-insulator transition of VO_2 on TiO_2 NM/Si.

Response to the reviewer's comments (Reviewer #3)

➤ Comment 1

Lee et al. report on monocrystalline nanomembranes (NM) of TiO₂ and its transfer to silicon substrates for use as template for VO₂ heteroepitaxy. The VO₂ films show very sharp M-I transitions. The main novelty of the paper is the use of VO₂ sacrificial layers (SL), deposited epitaxially on TiO₂ substrates and removed by a H₂O₂ solution. Compared to the popular Sr₃Al₂O₆ (SAO), VO₂ SL presents a lower lattice mismatch with TiO₂, expanding the range of epitaxial oxides that can be prepared as NM by etching of oxide SLs. In addition, VO₂ is much stable to air exposure than SAO, thus overcoming an important practical limitation of SAO.

The results increase the current capabilities of oxide NMs and thus I consider the results to be relevant. However, before recommending publication in Nature Communications, the authors should review the paper considering several points.

Responses: We thank the reviewer #3 for his/her positive comments on our work. We are pleased for him/her to recognize the novelty of our manuscript for the publishing in Nature Communications. We appreciate the reviewer #3 for his/her valuable time and effort in reviewing our manuscript. The careful comments and questions were very helpful to improve our manuscript. We have comprehensively revised our manuscript to provide more experimental data based on his/her insightful comments. Our responses are presented point by point in the following passages.

➤ **Comment 2**

1) In Introduction (p. 4, l. 75) the authors state "...the development of oxide NM has been limited for perovskite structure so far; to extend the materials spectrum for freestanding oxide NM, new combination of sacrificial layer and etchant needs to be developed..."

However, the recent paper cited in ref. 1 (Kum, H. S. et al. Heterogeneous integration of single-crystalline complex-oxide membranes. *Nature* 578, 75-81, (2020)) shows that oxide heteroepitaxy is possible when oxide substrates are covered by a few monolayers of graphene. In this reference, high quality oxide NM of perovskites, spinels and garnets are fabricated by this remote epitaxy method. Please revise the written sentence.

Response: We appreciate the reviewer #3 for the clarifying comments. Following the reviewer #3's suggestion, we have revised the sentence in our manuscript.

We also recognized that high quality oxide NM (perovskites, spinels and garnets, not rutile) are fabricated by this remote epitaxy method as reported in the referred literature from reviewer #3 (*Nature* 578, 75-81 (2020)). However, we would like to emphasize that remote epitaxy approach for oxide NM may not be suitable for synthesizing ultrathin oxide NM. To prevent oxidation of interfacial graphene to production of oxides membranes, epitaxial oxide films should be grown under vacuum ($< 5 \times 10^{-6}$ torr) at initial growth (about 5 – 10 nm) (*Nature* 578, 75-81 (2020)). Oxide film growth under vacuum may generate oxygen vacancies in the oxide films and/or interrupt accurate control of stoichiometric ratio (e.g., the formation of oxygen deficient sub-oxide); the generation of defects in ultrathin regime near the interface could suppress the intrinsic properties of multi-functional oxide films. Moreover, the coalescence of localized nuclei in oxide layers on the graphene, along with restriction of oxygen environment during growth, prevents the layer-by-layer growth of VO₂ NM with atomic precision and high quality using remote epitaxy.

Recently, remote epitaxy of rutile VO₂ NM was demonstrated by using graphene interfacial layer in another literature (*Nano Lett.* 20, 33-42 (2020)), but VO₂ NM exhibited domain boundaries by rotational symmetry mismatch between monoclinic (100)_R-VO₂ films with a

2/m point group and rhombohedral (0001) Al_2O_3 substrates with a $\bar{3}m$ point group. The existence of these extended defects prohibits steep metal-insulator transition of VO_2 NM by remote epitaxy.

By contrast, the freestanding NM was chemically released from the substrate by selective etching of sacrificial layers (i.e., chemical lift-off). Recently, atomically thin perovskite oxide membrane was gently released by dissolving water-soluble $\text{Sr}_3\text{Al}_2\text{O}_6$ sacrificial layers. But, the development of oxide NM has been limited for perovskite structure among chemical lift-off methods so far.

In these respects, our approach for single-crystalline rutile oxide NM are unique and shows some advantage in some aspect compared to remote epitaxy; our approach would be generally applicable for heterogeneous integration of oxide NM with any rutile crystal structures on Si substrates.

Changes (Page 4 in the manuscript): Following the reviewer #3's suggestion, we have revised the sentence in the Introduction of the manuscript as "the development of oxide NM has been limited for perovskite structure among chemical lift-off methods so far".

➤ **Comment 3**

2) XRD scans shown in the manuscript were measured using synchrotron radiation. Similar scans would be useful, at least some of them presented as Supplementary Information, (for example to other authors who may try to replicate the fabrication method in the future).

Response: We thank reviewer #3 for very constructive suggestion to improve our manuscript. Before synchrotron high-resolution x-ray scattering measurement, we also measured heterogeneous integration of single-crystalline TiO₂ NM on silicon substrate using in-house HRXRD with Cu K_{α1} radiation.

Changes (Page 7 in the manuscript, SI): We revised the sentence in the manuscript and added a figure (**Fig. R11**) to the **Supplementary Figure 5**.

Figure R11 X-ray diffraction 2θ - ω scan of transferred TiO₂ NM on silicon substrate using in-house HRXRD with Cu K_{α1} radiation

➤ **Comment 4**

3) The VO₂ SL is deposited at 300 °C. Please indicate, or confirm, if the deposition conditions for VO₂ on the transferred TiO₂ NM are the same.

Response: We apologize the lack of our explanation and thank the reviewer #3 for missing information of experimental details. Obviously, this information should be included in the manuscript. The epitaxial VO₂ film was successfully grown on the TiO₂ NM/Si with an identical growth condition of that of VO₂ sacrificial layers on the (001) TiO₂ substrate.

Changes (Page 15 in the manuscript): Based on the reviewer #3's suggestions, we added the sentence in the Method Section that the growth conditions for VO₂ films on TiO₂ NM/Si is identical with that for VO₂ sacrificial layer on TiO₂ substrates before the release of NM.

➤ **Comment 5**

4) The indicated area of the NM is "a few millimeters". Can be obtained and transferred NMs of larger area (at least the largest areas usual in standard PLD, around 10x10 or 10x8 mm²)? How does the etching time depend on the area?

Response: The reviewer #3 asked the achievable area of NMs and the area dependence of etching time for the removal of VO₂ sacrificial layers.

As the reviewer #3 commented, the lateral size of NMs is determined by the size of TiO₂ substrates, as long as laterally uniform growth is allowed at optimized condition of PLD. Since the laser-induced plasma plume could be uniformly distributed for up to 5 mm × 5 mm TiO₂ substrates in the optimized growth condition, the largest lateral area of NMs was 5 mm × 5 mm in our case. However, the size could be further increased in case of the increased substrate size and larger area deposition technique (e.g., sputtering or MOCVD).

On the area dependence of etching time for the removal of VO₂ sacrificial layers, we estimated the release time of TiO₂ NMs attached on supporting layer from as-grown TiO₂ (70 nm)/VO₂ (15 nm)/TiO₂ heterostructures with various lateral size (1 mm × 1 mm, 2.5 mm × 2.5 mm, 5 mm × 5 mm); as-grown TiO₂/VO₂/TiO₂ was immersed in dilute H₂O₂ solution (50 ml H₂O₂ + 150 ml H₂O). Since un-reacted TiO₂ epitaxial layers cover whole surfaces of VO₂ sacrificial layers during release process, release of epitaxial TiO₂ NM by oxidation and dissolution of VO₂ sacrificial layers begins from the edge of the substrate (i.e., side etching). Therefore, the etching time increases in proportion to the size of the substrate as shown in **Fig. R12** (~ 6 hours, 10 hours and 22 hours for 1 mm × 1 mm, 2.5 mm × 2.5 mm and 5 mm × 5 mm, respectively). This etching rate of heterostructure with TiO₂ upper blanket is much slower than naked VO₂ sacrificial layer in the **Supplementary Figure 4** of the manuscript. However, considering etching time of previously reported Sr₃Al₂O₆ sacrificial layer, which take ~ 1 day to delaminate SrTiO₃ and YBa₂Cu₃O_{7-x} NM (1 mm × 2.5 mm) using water (pH ~ 7) or potassium hydroxide (pH ~ 12) etchants (*Nat. Mater.*, 15, 1255-1260, (2016) & *Phys. Rev. Mater.*, 3, 060801, (2019)), the combination of VO₂ sacrificial layer and H₂O₂ etchant shows

a more than twice faster etch rate.

Changes (Page 7, 15 in the manuscript and SI): we added some sentences on the achievable area of NMs and the area dependence of etching time for the removal of VO₂ sacrificial layers in the manuscript and the related figures in the **Supplementary Figure 4**.

Figure R12 The release time of TiO₂ NMs attached on supporting layers from as-grown TiO₂ (70 nm)/VO₂ (15 nm)/TiO₂ heterostructures with various sample size (1 mm × 1 mm, 2.5 mm × 2.5 mm, 5 mm × 5 mm)

➤ **Comment 6**

5) Can the authors provide more information/results on the mechanical stability of the transferred TiO₂ NMs? The mechanical stability could be limited by stress at the VO₂/TiO₂ interface (f is around 0.86%) or at the TiO₂/Si interface (TEC mismatch).

5A) Are VO₂ films tens of nm thick on TiO₂/Si stable?

[REDACTED]

[REDACTED]

[REDACTED]

Figure R15 **a.** repeated temperature-dependent resistance measurement from 270 K to 320 K during 5 cycles. **b.** thermal cycling at 278 K and 308 K until 100 cycles in the 10 nm-thick VO₂ / 70 nm-thick TiO₂ NM.

Despite the interesting phenomena on the bendable NM by misfit strain between 70-nm-thick VO₂ and TiO₂, we would like to emphasize that 10 nm-thick VO₂ film grown on the 70 nm-thick TiO₂ NM is mechanically stable even after repetitive thermal cycling. We performed the repeated thermal cycling by measuring temperature-dependent sheet resistance at several times (270 K ~ 320 K). Obviously, steep sheet resistance modulation with metal-insulator transition across $T_{MI} \sim 296$ K ($\Delta\rho/\rho \sim 10^3$) were observed in 10 nm-thick VO₂ on TiO₂ NM /Si more than 100 cycles (**Fig. R15**); this result confirms that 10 nm-thick VO₂ on TiO₂ on the 70 nm-thick TiO₂ NM mechanically robust without degradation.

Changes (Page 12 in the manuscript and SI): We hope we fully addressed reviewer #3's comment on the mechanical stability of VO₂ on TiO₂ NM, and we also added these sentence in the revised manuscript and figures in the **Supplementary Figure 14**.

➤ **Comment 7**

5B) Are films (VO₂ or other oxides) deposited at higher temperature (example 700 °C) stable?

Response: We appreciate the reviewer #3 for constructive suggestion. First of all, VO₂ film was not suitable to high growth temperature (> 500 °C) due to their low melting point. Instead, RuO₂ films were grown on TiO₂ NM / Si substrate by pulsed laser deposition from sintered RuO₂ target at higher substrate temperature (T_G ~ 600 °C) due to more tolerance to high temperature (Adv. Mater. **20**, 501-505 (2008)). Symmetric 2θ-ω scan using synchrotron x-ray scattering detected epitaxial (002) RuO₂ peak with (002) TiO₂ and (400) Si peak (**Fig. R16**). The peak position of (002) RuO₂ on TiO₂ NM/Si is identical with that of (002) RuO₂ on TiO₂ substrates; this result indicates that RuO₂ film can be epitaxially grown at 600°C since TiO₂ NM is stable without degradation at higher temperature even under oxidizing atmosphere, where graphene interlayer is not stable for remote epitaxy.

Changes (Page 14 in the manuscript): We hope we fully addressed reviewer #3's comment and we also added the sentences on the thermal stability of single-crystalline TiO₂ NM.

Figure R16 Schematic of **a.** RuO₂ film grown on the TiO₂ NM / Si substrate and **c.** RuO₂ film grown on the (001) TiO₂ substrate at substrate temperature of 600 °C and corresponding symmetric 2θ-ω scans of **b.** RuO₂ / TiO₂ NM / Si and **d.** RuO₂ / (001) TiO₂ substrate. The location of (002) peak for RuO₂ films grown at 600 °C on TiO₂ NM/Si was almost identical that of epitaxial RuO₂ films grown at 600 °C on (001) TiO₂ substrates.

➤ **Comment 8**

6) Can another oxide, different from TiO₂, be deposited on VO₂ SL, or can it be deposited on TiO₂ / VO₂ before etching?

Response: We thank reviewer #3 for clarifying and constructive comments. The reviewer #3 asked if another oxide, different from TiO₂, could be grown on the VO₂/TiO₂ heterostructure before the removal of VO₂ sacrificial layers (SL). Of course, we can absolutely grow other “epitaxial” oxides with rutile structures, and this oxide layers can be released and transferred on Si substrates, as long as the oxide are insoluble under dilute H₂O₂ solution; Among many candidates of rutile oxides, RuO₂ NM was selected to prove the generality of our approach for “single-crystalline” rutile oxide NM.

Following the reviewer #3’s suggestion, we firstly grew 9 nm-thick RuO₂ films on the VO₂/TiO₂ heterostructure under $pO_2 = 40$ mTorr and $T_g = 400$ °C (**Fig. R17 a**). Then, we performed symmetric XRD 2 θ - ω scans of RuO₂/VO₂ on TiO₂ substrates (**Fig. R17 e**). As expected, strong (002) RuO₂ peak appeared at ~ 47.8 °. Then, rigid supporting layer was coated on the RuO₂/VO₂/TiO₂ heterostructure and immersed it into H₂O₂ solution at room temperature; RuO₂ nanomembranes were released from the TiO₂ substrates by using selective etching of VO₂ sacrificial layer (**Fig. R17 b**). After fully dissolving VO₂ sacrificial template, RuO₂ nanomembranes was released, and then transferred on Si substrate (**Fig. R17 c**), followed by the removal of rigid supporting layer (**Fig. R17 d**). Symmetric 2 θ - ω scan by using synchrotron XRD showed that (002) RuO₂ peak appeared with (400) Si substrate (**Fig. R17 e**), which confirms freestanding epitaxial RuO₂ film was successfully transferred on the Si substrate.

Figure R17 a. schematic of an epitaxial RuO₂/VO₂ heterostructure on TiO₂ mother substrate. **b,** The VO₂ layer is dissolved in H₂O₂ to release the top RuO₂ film with the mechanical supporting layer (e.g., PDMS and TRT). **c,** The freestanding RuO₂ NM is transferred onto the Si substrate **d,** By removing the rigid supporting layer, single-crystalline RuO₂ is heterogeneously integrated into a silicon substrate. Corresponded symmetric 2θ-ω XRD scans of the **e.** Fig. R6 a and **f.** Fig. R6 d. The epitaxial RuO₂ film was successfully transferred on the Si substrate.

Changes (Figure 14 in the manuscript and SI): We added some sentences in the revised manuscript and related figures in the **Supplementary Figure 20** on the general application of our method for the production of “single-crystalline” rutile oxide nanomembranes.

➤ **Comment 9**

7) Can the authors illustrate the potential of the TiO₂ NM by depositing other functional oxide(s)?

Response: We thanks the reviewer #3 for very constructive suggestion. The reviewer #3 suggested that the “general” potential of our approach for the production of TiO₂ NM needs to be illustrated by depositing other functional oxides, as well as VO₂, which is the essentially same suggestion with reviewer #1. We would like to emphasize that our work is generally applicable for the fabrication of single-crystalline rutile oxide nanomembranes.

Figure R18 **a**, schematic of an epitaxial TiO₂/VO₂ heterostructure on TiO₂ mother substrate. **b**, The VO₂ layer is dissolved in H₂O₂ to release the top TiO₂ film with the mechanical supporting layer (e.g., PDMS and TRT). **c**, The freestanding TiO₂ NM is transferred onto the Si substrate. **d**, By removing the rigid supporting layer, single-crystalline TiO₂ NM is heterogeneously integrated into a silicon substrate. **e**, epitaxial RuO₂ film is grown on the TiO₂-NM-templated Si substrates.

To generally demonstrate the heterogeneous integration of other rutile oxide single-crystalline on Si, we grew epitaxial RuO₂ films on TiO₂ NM/Si (**Fig. R18**). After the cleaning process to remove the residue on transferred TiO₂ NM, 8.5-nm-thick RuO₂ thin films, instead of VO₂ films, were grown on TiO₂ NM/Si by pulsed laser deposition (**Fig. R18e**). As observed in symmetrical XRD 2 θ - ω scans, the intense (002) RuO₂ peak appeared at $\sim 2\theta = 47.8^\circ$ (**Fig.**

R19 b), along with peaks related to the TiO₂ template ($\sim 2\theta = 49.5^\circ$) and Si substrates ($\sim 2\theta = 54.3^\circ$). The location of (002) RuO₂ peak on TiO₂ NM/Si was almost identical that of epitaxial RuO₂ films grown on (001) TiO₂ substrates (**Fig. R19 d**); this result implicates that TiO₂ NM templates facilitate the formation of epitaxial RuO₂ films on Si substrates as well.

Figure R19 Schematic of **a.** RuO₂ film grown on the TiO₂ NM / Si substrate and **c.** RuO₂ film grown on the (001) TiO₂ substrate and corresponding symmetric 2θ - ω XRD scans of **b.** RuO₂ / TiO₂ NM / Si and **d.** RuO₂ / (001) TiO₂ substrate. The location of (002) RuO₂ peak on TiO₂ NM/Si was almost identical that of epitaxial RuO₂ films grown on (001) TiO₂ substrates.

The cross sectional HAADF-STEM images confirm that the epitaxial growth of 9-nm-thick RuO₂ on TiO₂ NM can realize heterogeneous integration of single-crystal RuO₂ films on a Si substrates. The RuO₂ film appears much brighter than TiO₂ NM because the atomic number of Ru ($Z_{\text{Ru}} = 44$) is much larger than that of Ti ($Z_{\text{Ti}} = 22$). The low magnification image reveals

that the RuO₂ thin films were uniformly grown on TiO₂ NM (**Fig. R20 a**). The atomic-scale resolution image near the interface between RuO₂/TiO₂, indicated by yellow square in **a**, is shown in **Fig. RX 20 b** (zone axis : [100] in both TiO₂ and RuO₂). The atomic columns of RuO₂ and TiO₂ are coherently matched at the interface with perfect rutile crystal without any noticeable defects; it means TiO₂ NM on Si allows the epitaxial growth of RuO₂ films; high-magnification HAADF-STEM image shows perfect registry of Ru atoms in the rutile structure and sharp interface from the contrast difference between RuO₂ and TiO₂. Selective area electron diffraction (SADP) spots on RuO₂/TiO₂ NM regions show that both RuO₂ film and TiO₂ NM have high quality of rutile crystal structures (**Fig. R20 c**). The out-of-plane direction diffraction spots (i.e., 002, indicated by blue square) are slightly separated due to lattice mismatch between RuO₂ and TiO₂, whereas the in-plane direction diffraction spots (i.e., 040, indicated by red square) are completely overlapped as a single spot, also confirming that epitaxy growth of RuO₂ thin films on TiO₂ NM. When comparing the lattice parameter c/a ($=c/b$) ratio calculated from the SADP images with the c/a ratio in the bulk states, there were differences by -0.98% and -4.48% in TiO₂ and RuO₂, respectively. The large decrease of c/a in RuO₂ indicates that the RuO₂ films are fully strained by biaxial tensile strain along the in-plane direction; SADP results confirmed perfect epitaxy between TiO₂ NM and RuO₂ layers transferred on Si, and thus TiO₂ NM templates between RuO₂ and Si enable epitaxial growth of RuO₂ films, as well as VO₂ films, with crystallographic perfection free from defects.

Figure R20 STEM analysis of RuO₂/TiO₂ NM on SiO₂/Si substrate. **a**, The cross sectional HAADF-STEM image of the heterogeneous RuO₂/TiO₂ NM on SiO₂/Si. The RuO₂ film appears much brighter than TiO₂ NM because the atomic number of Ru ($Z_{\text{Ru}}=44$) is much larger than that of Ti ($Z_{\text{Ti}}=22$). The low magnification image reveals that the RuO₂ thin films were uniformly deposited on TiO₂ NM with 8.5 nm thickness. The atomic-scale resolution image near the interface

between RuO₂/TiO₂, indicated by yellow square in **a**, is shown in **b** (zone axis : [100] in both TiO₂ and RuO₂). The atomic columns of RuO₂ and TiO₂ are coherently matched at the interface with perfect rutile crystal structure; it means TiO₂ NM on SiO₂/Si allows the epitaxy growth of RuO₂ thin films despite the large lattice mismatch (~ 2%). **c**. SADP images on RuO₂/TiO₂ NM regions. The sharp diffraction spots shows both RuO₂ film and TiO₂ NM have high quality of rutile crystal structures. The out-of-plane direction diffraction spots (i.e., 002, indicated by blue square) are slightly separated due to lattice mismatch between RuO₂ and TiO₂, whereas the in-plane direction diffraction spots (i.e., 040, indicated by red square) are completely overlapped as a single spot, also confirming that epitaxy growth of RuO₂ thin films on TiO₂ NM. When comparing the lattice parameter $c/a(=c/b)$ ratio calculated from the SADP images with the c/a ratio in the bulk states, there were differences by -0.98% and -4.48% in TiO₂ and RuO₂, respectively. The large decrease of c/a in RuO₂ indicates that the RuO₂ films are fully strained by biaxial tensile strain along the in-plane direction.

Besides, to experimentally verify the local strain analysis of RuO₂ / TiO₂ NM / Si substrates, we performed the geometric phase analysis (GPA) strain quantification using STEM techniques (Fig. R21 a). In-plane (ϵ_{IP}) and out-of-plane (ϵ_{OOP}) lattice strain mapping were obtained from geometric phase analysis (GPA) of HAADF-STEM image. Line profile of lattice strain, ϵ_{IP} and ϵ_{OOP} , on the RuO₂ films were extracted from GPA strain mapping (blue box in Fig. R21 b). The lattice strains were calculated based on the lattice parameter of reference region in TiO₂ NM (yellow box in Fig. R21 a). In-plane lattice parameters were almost invariable across the RuO₂/TiO₂ NM (Fig. R21 b, c), indicating that the RuO₂ thin films were completely constrained from the TiO₂ NM-temple. On the other hands, out-of-plane lattice parameters of RuO₂ films were about 4.5% larger than that of TiO₂ NM (Fig. R21 b, c), and these are also uniform over the entire area of the films.

From our results, we clearly shows that “single crystalline” TiO₂ NM had promising template to integrate other “single-crystalline” functional oxide with rutile structure on Si substrates.

Changes (Page 14 in the manuscript and SI): We hope that we fully addressed reviewer #3’s comment about “general” potential of TiO₂ NM on our manuscript and we also added a sentence in the revised manuscript and related figures in **Supplementary Figures 17-19**.

Figure R21 Local strain analysis of RuO₂/TiO₂ NM on SiO₂/Si substrate. **a**, HAADF-STEM image of RuO₂ thin films on the TiO₂ NM-template. **b**, In-plane (ϵ_{IP}) and out-of-plane (ϵ_{OOP}) lattice strain mapping obtained from geometric phase analysis (GPA) of HAADF-STEM image **a**. **c**, Line profile of lattice strain, ϵ_{IP} and ϵ_{OOP} , on the RuO₂ films extracted from GPA strain mapping **b** (blue box). These lattice strains were calculated based on the lattice parameter of reference region in TiO₂ NM (yellow box in **a**). The interface between RuO₂/TiO₂ NM is identified by the orange dotted lines in **a**, **b**, and **c**. In-plane lattice parameters almost invariable across the RuO₂/TiO₂ NM, indicating that the RuO₂ thin films were completely constrained from the TiO₂ NM-template. On the other hands, out-of-plane lattice parameters of RuO₂ films are about 4.5% larger than that of TiO₂ NM, and these are also uniform over the entire area of the films. It is slightly different with bulk state, where the gap of out-of-plane lattice parameter between RuO₂ and TiO₂ is about 5.4%, because the RuO₂ films were tensile-strained along the in-plane direction.

➤ **Comment 10**

8) The authors compare the properties of the VO₂ film deposited on the transferred NM to VO₂ films grown on SiO₂/Si. The films on SiO₂/Si are polycrystalline and thus the enhancement of properties is not surprising. Can it be compared to VO₂ films grown epitaxially directly on buffered Si(001)?

Responses and Changes: We appreciate the reviewer #3 for very constructive suggestion. As the reviewer #3 commented, a number of buffer layers (e.g., AlN, YSZ, Al₂O₃,) have been utilized for epitaxial VO₂ films on (001) Si to improve a crystal quality and metal-insulator transition characteristics of the VO₂ film in the previous literatures (e.g., *APL Mater.* **4**, 026101 (2016), *Appl. Phys. Lett.* **95**, 111915 (2009), *Jpn. J. Appl. Phys.* **47**, 3067 (2008), *Nano-Micro Lett.*, **9**, 29 (2017), *J. Phys. D: Appl. Phys.*, **47**, 455304 (2014), *J. Appl. Phys.* **127**, 205303 (2020)), but all films could be grown by domain match epitaxy, not layer-by-layer epitaxy; these films exhibit fundamentally inferior structural and electrical quality, especially near the interfaces between VO₂ films and buffer layers. For instance, 70 nm thick-VO₂ film was grown on the yttria-stabilized zirconia (YSZ) templated Si substrate with following crystallographic relationship (VO₂ (010)[100] || YSZ (001)[110] || Si (001)[100]) (*Appl. Phys. Lett.* **95**, 111915 (2009)). As another example, 50 – 120 nm thick VO₂ films were epitaxially grown on the AlN templated Si substrate having hexagonal domains (*APL Mater.* **4**, 026101 (2016)). In addition to domain matching epitaxy with lattice matched buffer layer, Al₂O₃ buffer layer could improve preferential orientation of the VO₂ film; XRD scans of VO₂/Al₂O₃/Si (001) substrate indicated that the Al₂O₃ buffer layer facilitate formation of stoichiometric VO₂ thin film along with [011] crystal orientation on Si substrate (*Nano-Micro Lett.*, **9**, 29 (2017), *J. Phys. D: Appl. Phys.*, **47**, 455304 (2014)).

Despite significant efforts to improve structural and electrical characteristics of VO₂ thin films using the above buffer layers on silicon, we would like to emphasize that high resistivity modulation ($\Delta\rho/\rho$) is difficult to be achieved in ultrathin regime of VO₂ thickness due to fundamental limitation of this approach. As shown in previously reported literatures (*APL Mater.* **4**, 026101 (2016)), the degree of high resistivity modulation ($\Delta\rho/\rho$) across the metal-

insulator transition was strongly suppressed as VO₂ thickness decreased from 130 nm to 50 nm. (No temperature-dependent MIT result was reported in thinner VO₂ films (< 50 nm) on AlN-buffered Si, probably due to inferior $\Delta\rho/\rho$ modulation). Indeed, we grew VO₂ films on the Al₂O₃ buffered Si substrate with various thickness (10 nm ~ 150 nm) for a direct comparison. The $\Delta\rho/\rho$ in 10-nm-thick VO₂ films ($\Delta\rho/\rho_{10nm} \sim 1.50$) were considerably reduced compared with that of 150-nm-thick VO₂ film on Al₂O₃-buffered/Si substrate ($\Delta\rho/\rho_{150nm} \sim 4.80 \times 10^2$) (**Fig. R22a**); The $\Delta\rho/\rho$ in 10-nm-thick “single-crystalline” VO₂ films on TiO₂ NM/Si ($\Delta\rho/\rho \sim 3.3 \times 10^3$) shows more than 3 orders of magnitude higher than that in 10-nm-thick VO₂ films on Al₂O₃-buffered/Si substrate ($\Delta\rho/\rho_{10nm} \sim 1.50$). In the case of VO₂ films on buffered Si, a further reduction in $\Delta\rho/\rho$ was found to occur with decreasing film thickness, likely due to a number of interfacial defects and domain boundaries between buffer layer and VO₂ film; By fair comparison with buffered Si based on reviewer #3’s suggestion, our work clearly demonstrates that exceptional MIT characteristics in terms of transition sharpness ($\Delta T_h \sim 2.85$ K, $\Delta T_c \sim 3.3$ K) and resistivity ratio ($\Delta\rho/\rho > 10^3$) were realized in ultrathin VO₂ films integrated on Si substrates, benefitting from the superior quality of epitaxial oxide films.

Changes (Page 12 in the manuscript and SI): We hope that we fully addressed reviewer #3’s comment on exceptional properties of the VO₂ epitaxial layers grown on the TiO₂ NM/Si, even compared to buffered Si substrates, on the revised manuscript. To demonstrate more clearly, we also added a sentence in the revised manuscript and a figure in **Supplementary Figure 16**.

Figure R22 a. temperature-dependent resistivity modulation near T_{MIT} in 10-nm-thick VO₂ film (gray line), 50-nm-thick VO₂ film (red line) and 150-nm-thick VO₂ film (blue line) grown on Al₂O₃ buffered Si substrate. **b.** Benchmark of resistivity ratio $\Delta\rho/\rho = (\rho_{T_{MIT}-15K} - \rho_{T_{MIT}+15K})/\rho_{T_{MIT}+15K}$ for VO₂ films on the buffered Si substrates across the MIT. For a direct comparison, all films were grown on the buffered Si substrates using various buffer layers: (Al₂O₃ buffer layer (green triangle) from ref. *Jpn. J. Appl. Phys.* **47**, 3067 (2008), *J. Phys. D: Appl. Phys.*, **47**, 455304 (2014), *J. Appl. Phys.* **127**, 205303 (2020), AlN buffer layer (black square) from ref. *APL Mater.* **4**, 026101 (2016), YSZ buffer layer (blue circle) from ref. *Appl. Phys. Lett.* **95**, 111915 (2009), TiN/Ti buffer layer (pink inverted triangle) from ref. *J. Vac. Sci. Technol. A*, **32**, 041502 (2014) and the Al₂O₃ buffer layer from **Fig. R22a** (orange diamond)).

REVIEWER COMMENTS

Reviewer #1 (Remarks to the Author):

I am happy with the responses to my comments. I am also amazed that the authors have further demonstrated heterogeneous integration of single-crystalline RuO₂ films on TiO₂. I suggest publishing this manuscript now.

There is one minor comment about the title. The current title is too general. The authors are suggested to add VO₂ into the title so it is more accurate.

Reviewer #2 (Remarks to the Author):

I studied all the referee comments and the revised manuscript. I found the comments are quite constructive. The authors have provided a detailed response to much of the comments and clarified what is the novel aspect of their work. The structural data provided is detailed to show the quality of their film structures. There are a few aspects outstanding from reading their response letter and revision file:

- The authors claim that the TiO₂ layer is essential to make this whole membrane structure. Clearly this is interesting, but also severely limits the practical value of this method. This basically means no out-of-plane use is possible which might limit any benefit of making a suspended membrane in the first place. The authors should clearly state this limitation in their manuscript to make fair claims on the approach.

- The authors show only one data point in figure 5 from their samples. They should populate that plot with at least 2-3 more points (3 points needed to make a line) to show how the resistance change scales with thickness. Otherwise it is not clear whether this method is generally relevant except for the magic thickness of 10nm.

- The authors claim the thermal resistance transition is very sharp. This is certainly interesting. The authors should add to this the sharpness of the electrical transition (IV jump) and whether this is any different from literature. If it is the same, then it will be interesting to discuss why this result is not any different? This data will be useful for the reader to understand whether this method is of particular use to VO₂ devices since there are already numerous approaches to make suspended VO₂ devices with very sharp electrical IMT behavior.

Reviewer #3 (Remarks to the Author):

The authors have improved the manuscript and the revised version, and the revised Supplementary Information, present now important information that was not detailed in the first version.

I recommend publication in Nature Communications.

Only a short comment. I suggest the authors to revise Supplementary Figure 10, where they indicate "...Except height modulation by the slight bending due to the elastic supporting layer,..." Please note that height modulation can also be an AFM artifact that was not corrected by applying the usual flattening filter included in all AFM software. The authors can probably distinguish whether the modulation is instrumental or intrinsic by also scanning larger areas to see if the lateral dimension does not change.

Response to the reviewer's comments (Reviewer #1)

➤ Comment 1

I am happy with the responses to my comments. I am also amazed that the authors have further demonstrated heterogeneous integration of single-crystalline RuO₂ films on TiO₂. I suggest publishing this manuscript now.

There is one minor comment about the title. The current title is too general. The authors are suggested to add VO₂ into the title so it is more accurate

Responses: We appreciate the reviewer #1 for his/her compliment on our work and his/her suggestion in publishing our manuscript to Nature Communications. We appreciate the reviewer #1 for his/her valuable time and effort in reviewing our manuscript.

Following the reviewer #1's comments related to too generalized title, we added "rutile" in the revised title to be more specific because we demonstrated both single-crystalline VO₂ and RuO₂ films on silicon. Thus, the title is now revised as "Heterogeneous integration of single-crystalline rutile oxide nanomembrane with steep phase transition on silicon substrates"

Response to the reviewer's comments (Reviewer #2)

➤ Comment 1

I studied all the referee comments and the revised manuscript. I found the comments are quite constructive. The authors have provided a detailed response to much of the comments and clarified what is the novel aspect of their work. The structural data provided is detailed to show the quality of their film structures. There are a few aspects outstanding from reading their response letter and revision file:

Response: We appreciate the reviewer #2 for his/her effort in reviewing our manuscript. We are pleased to deliver the novel aspect of our work from the previous revision. We revised a few remaining aspects that the reviewer #2 pointed out in this round.

➤ **Comment 2**

The authors claim that the TiO₂ layer is essential to make this whole membrane structure. Clearly this is interesting, but also severely limits the practical value of this method. This basically means no out-of-plane use is possible which might limit any benefit of making a suspended membrane in the first place. The authors should clearly state this limitation in their manuscript to make fair claims on the approach.

[REDACTED]

[REDACTED]

➤ **Comment 3**

The authors show only one data point in figure 5 from their samples. They should populate that plot with at least 2-3 more points (3 points needed to make a line) to show how the resistance change scales with thickness. Otherwise it is not clear whether this method is generally relevant except for the magic thickness of 10 nm.

Responses: We thank the reviewer #2 for the comment. As the reviewer #2 suggested, we added more data point with different thickness of VO₂ layer in Figure 5. Of course, 10 nm is not magic thickness; different thickness (i.e., 5 nm in this additional experiment) of VO₂ films on TiO₂ NM/Si also show sharp metal-insulator transition characteristic with high resistivity ratio ($\Delta\rho/\rho > 10^3$) (**Fig. R2 a**), as long as the thickness of VO₂ layers are below the critical thickness (~ 15 nm) for micro-crack generation. Since no discern difference was observed depending on the thickness of VO₂ layers, further point is not necessary to make a line. Obviously, this is astonishing result considering the complete degradation of metal-insulator transition ($\Delta\rho/\rho \sim 10$) of 5-nm thick VO₂ films on silicon substrates. (see *Nano Lett.* **17**, 1762-1767 (2017), *Adv. Funct. Mater.* **25**, 679-686 (2015)). (**Fig. R2 b**)

Changes: We added more data in Figure 5 and changed linear scale in x-axis into log scale in x-axis to distinguish the data in our work. We also added more sentences in the revised manuscript and temperature-dependent resistivity data for 5 nm-thick VO₂ films on TiO₂ NM/Si in Supplementary Figure 13b.

Figure R1 | **a**, Temperature-dependent resistivity from 270 K to 320 K in the 5 nm-thick VO₂ films on 50 nm-thick TiO₂ NM/Si. **b**, Benchmark of resistivity ratio $\Delta\rho/\rho = (\rho_{T_{MIT}-15K} - \rho_{T_{MIT}+15K})/\rho_{T_{MIT}+15K}$ for VO₂ films on the Si substrates across the MIT. (replaced Figure 5)

➤ **Comment 4**

The authors claim the thermal resistance transition is very sharp. This is certainly interesting. The authors should add to this the sharpness of the electrical transition (IV jump) and whether this is any different from literature. If it is the same, then it will be interesting to discuss why this result is not any different? This data will be useful for the reader to understand whether this method is of particular use to VO₂ devices since there are already numerous approaches to make suspended VO₂ devices with very sharp electrical IMT behavior.

Responses: Obviously, our two-terminal “single-crystalline” VO₂ threshold device on TiO₂ NM/Si shows higher I_{on}/I_{off} ratio (~ 10³) compared with other “defective” VO₂ threshold switching with lateral configuration in the previous literatures (I_{on}/I_{off} ~ 10, e.g., *New J. Phys.* 6, 52 (2004), *J. Appl. Phys.* 108, 073708 (2010)). This clearly proves the superior properties (i.e., electrically triggered metal-insulator transition) of our VO₂ films on TiO₂ NM/Si, which is originated from excellent structural quality.

Changes: We added some discussion on the superior properties (i.e., electrically triggered metal-insulator transition) of our VO₂ films on TiO₂ NM/Si in Supplementary Information.

Response to the reviewer's comments (Reviewer #3)

➤ Comment 1

The authors have improved the manuscript and the revised version, and the revised Supplementary Information, present now important information that was not detailed in the first version.

I recommend publication in Nature Communications.

Only a short comment. I suggest the authors to revise Supplementary Figure 10, where they indicate "...Except height modulation by the slight bending due to the elastic supporting layer,..." Please note that height modulation can also be an AFM artifact that was not corrected by applying the usual flattening filter included in all AFM software. The authors can probably distinguish whether the modulation is instrumental or intrinsic by also scanning larger areas to see if the lateral dimension does not change.

Responses: We thank the reviewer #3 for his/her constructive comments on our manuscript and the recommendation for the publication in Nature Communications.

To acquire the AFM image of released TiO₂ NM on supporting layer, we used soft supporting layer, which is easily bendable. This image was acquired after the application of the usual flattening filter included in AFM software; the height modulation is not AFM artifact

We also could distinguish whether the modulation is instrumental or intrinsic by scanning different spot. Contrary to the AFM image and depth profile in the original SI, which shows outward bending (**Fig. R3**), the other spot shows inward bending due to flexibility of organic supporting layer (**Fig. R4**). This result confirmed the slight bending signature of the TiO₂ NM on the supporting layer is intrinsic issue due to flexible supporting layer.

Changes: We added figure and sentences in Supplementary Figure 10.

Figure R3 | **a.** Atomic force microscopy image and **b.** depth profile along the x-axis direction of released TiO₂ NM on supporting layer. Except height modulation by the slight bending due to the elastic supporting layer, no VO₂ residue or any damages was observed at the surface of TiO₂ NM after H₂O₂ treatment.

Figure R4 | Inward bending in the other spot. this result confirmed the slight bending signature of the TiO₂ NM on the supporting layer is intrinsic issue due to flexible supporting layer.